# TOR and heat shock response pathways regulate peroxisome biogenesis during proteotoxic stress

Nandini Shukla [1,5] ✉, Maxwell L. Neal [2], Jean-Claude Farré[1], Fred D. Mast [2,3], Rajasri Sarkar[1], Linh Truong[1], Theresa Simon[1], Leslie R. Miller[2], John D. Aitchison [2,3,4] & Suresh Subramani [1] ✉

Peroxisomes are versatile organelles mediating energy homeostasis and redox balance. While peroxisome dysfunction is linked to numerous diseases, the mechanisms regulating peroxisome dynamics during cellular stress remain elusive. Using yeast, we show that proteotoxic stress, including loss of endoplasmic reticulum (ER) or cytosolic chaperone function, impaired ER protein translocation, disrupted N-linked glycosylation, or reductive stress, triggers peroxisome proliferation. This occurs through increased de novo biogenesis from the ER, as well as growth and division, rather than impaired pexophagy. Peroxisome biogenesis is essential for cellular recovery from proteotoxic stress. Through comprehensive testing of major signaling pathways, we determine this response to be mediated by activation of the heat shock response and inhibition of Target of Rapamycin (TOR) signaling. Notably, the effects of proteotoxic stress and TOR inhibition on peroxisomes are also observed in human fibroblasts. Our findings reveal a critical and conserved role of peroxisomes in cellular response to proteotoxic stress.

Peroxisomes are essential, evolutionarily conserved organelles that play key roles in energy metabolism via fatty acid oxidation and redox homeostasis[1]. Their biogenesis is controlled by *PEX* genes, many of which are conserved across species, and their turnover occurs by selective autophagy, known as pexophagy[2]. Peroxisome biogenesis occurs either by growth and division of pre-existing peroxisomes, or by de novo biogenesis, which involves the budding and subsequent fusion of pre-peroxisomal vesicles from the endoplasmic reticulum (ER)[2,3]. The balance between peroxisome biogenesis and turnover maintains its homeostasis, which is sensitive to nutritional cues. This is exemplified in single-celled yeasts by the induction of peroxisomes in cells grown in oleate because fatty acid oxidation is exclusively peroxisomal[4]. In rodents, a diverse class of compounds called peroxisome proliferators transcriptionally induce genes encoding fatty acid metabolizing enzymes, coincident with peroxisome proliferation[5]; however, the regulation of human peroxisomes remains unclear[6]. Not surprisingly, significant efforts have focused on the intracellular signaling pathways responding to nutritional cues resulting in peroxisome induction[4,7,8]. However, peroxisomes also respond to environmental stresses, but the studies investigating the signaling components and mechanisms involved are limited in both number and scope[9–11]. The importance of peroxisomes arises not only from their metabolic specialization, but also from their roles in immunometabolism, development, aging, and human disease[12–15], including human peroxisome biogenesis disorders (PBDs). This is the driving imperative to understand peroxisome biogenesis and its modulation, particularly its role in the adaptation to varying forms of abiotic environmental conditions, such as heat[10,16], light[17], pH, salt[11],

[1]Department of Molecular Biology, School of Biological Sciences, University of California, San Diego, La Jolla, CA, USA. [2]Center for Global Infectious Disease Research, Seattle Children's Research Institute, Seattle, WA, USA. [3]Department of Pediatrics, University of Washington, Seattle, WA, USA. [4]Department of Biochemistry, University of Washington, Seattle, WA, USA. [5]Present address: Department of Biological Sciences, University of North Texas, Denton, TX, USA. ✉e-mail: nandini.shukla@unt.edu; ssubramani@ucsd.edu

redox[18,19], and organelle stress[9], which necessitate a coordinated cellular response, often requiring contributions from several sub-cellular compartments and membrane contact sites[2,20–22].

Here, we demonstrate that proteotoxic ER stress induced by tunicamycin treatment causes peroxisome proliferation in *Saccharomyces cerevisiae*, *Komagataella phaffii*, and primary human fibroblasts. By manipulating the many arms of the ER stress response either genetically or pharmacologically, we discern that proteotoxic ER stress increases peroxisome number partially by activating the heat shock response (HSR) in the cytosol and by inhibiting TOR1. Mechanistically, this is executed via increased peroxisome production mainly through the de novo mode along with some fission of pre-existing peroxisomes, instead of impaired pexophagy. Importantly, peroxisome biogenesis is necessary for cell survival in response to tunicamycin treatment. Our study thus sheds light on the role of de novo peroxisome biogenesis in cellular adaptation to ER stress, with emphasis on evolutionary conservation of this adaptation, the role of multiple sub-cellular compartments in coordinating the response, as well as the types of stress and signaling pathways that activate this proliferation response. Overall, this study paves the path for a better understanding of peroxisome biogenesis in response to environmental insults, while also presenting potential applications in the management of PBDs.

## Results

### Inactivation of Kar2/BiP triggers UPR and causes peroxisome proliferation

While analyzing essential *S. cerevisiae* mutants for peroxisomal phenotypes, such as altered number using the fluorescent peroxisomal matrix marker, GFP-ePTS1, we observed that the temperature-sensitive mutant, *kar2-159*, exhibited a strikingly elevated number of peroxisomes compared to wild-type (WT). A more quantitative analysis of peroxisome number from 3D images performed using our recently developed software, perox-per-cell[23], revealed that both, the WT and the *kar2-159* mutant cells, showed a wide distribution in peroxisome number at permissive (25 °C) and restrictive (37 °C) temperatures. However, *kar2-159* cells consistently displayed significantly more peroxisomes at 37 °C, than at 25 °C. In comparison, peroxisome proliferation was significantly, but only mildly, induced in WT cells in response to the temperature shift (Fig. 1a, b).

Kar2, the yeast homolog of mammalian BiP, belongs to the Hsp70 family of chaperone proteins residing in the ER lumen. During protein misfolding stress in the ER lumen, Kar2 binds to unfolded proteins thereby dissociating from its binding partner, Ire1[24]. Inactivation of Kar2 induces the unfolded protein response (UPR, monitored by induction of UPRE-GFP) (Supplementary Fig. 1a, b), as well as the cytosolic heat shock response (HSR, monitored by induction of HSE-GFP) (Supplementary Fig. 1c, d), when compared to WT cells at 37 °C[24]. To understand how loss of Kar2 function is connected to peroxisome homeostasis, we investigated the effects of UPR and HSR activation on peroxisome proliferation.

### Induction of proteotoxic ER stress causes peroxisome proliferation and upregulates peroxisomal 3-ketoacyl-CoA thiolase

After confirming the induction of UPRE-GFP by tunicamycin (Supplementary Fig. 1e, f) (Tm), a nucleoside antibiotic that interferes with N-glycosylation of proteins in the ER lumen, we tested its effect on peroxisome number by counting Pex3-GFP and GFP-ePTS1 puncta. Pex3 is a peroxisomal membrane protein (PMP), whereas GFP-ePTS1 is imported into the peroxisome lumen. Tunicamycin (non-lethal doses, Supplementary Fig. 1i) significantly increased the number of peroxisomes as visualized by both markers (Fig. 1c–f), as well as the cumulative area occupied by all peroxisomes per cell (Supplementary Fig. 1j, k), within 2–5 h of treatment (Supplementary Fig. 1n). We also verified that all GFP-ePTS1 and Pex3-GFP puncta represent bona fide peroxisomes in both DMSO and tunicamycin-treatment conditions

(Supplementary Fig. 1l, m), and that ER stress induced by non-lethal doses of DTT also increased peroxisome number (Fig. 1g and Supplementary Fig. 1i). We confirmed a recent report, which suggested that peroxisome numbers (monitored indirectly by total Pex11-mNeonGreen fluorescence)[9] increase as an adaptation to supply additional acetyl-CoA via peroxisomal β-oxidation to support mitochondrial respiration during ER stress. To directly explore this idea, we examined whether ER stress induced peroxisomal β-oxidation enzymes, which are repressed during growth in glucose. Endogenous levels of GFP-tagged 3-ketoacyl-CoA thiolase (Pot1)[8], which catalyzes the final step in peroxisomal β-oxidation, increased following tunicamycin treatment (Fig. 1h, i). Collectively, our observations indicate that ER stress resulting from misfolded proteins, caused either by loss of Kar2 function or by tunicamycin, results in peroxisome proliferation (Fig. 1j and Supplementary Fig. 1).

To investigate if UPR induction causes peroxisome proliferation, we abrogated the UPR by either deleting *IRE1* or *HAC1*, the latter encoding the primary transcription factor mediating the UPR transcriptional response in yeast. Notably, the transcription factor, Gcn4, is also upregulated by ER stress, binds to Unfolded Protein Response Elements (UPREs), and is required for the induction of several UPR target genes, including *PEX* genes (*PEX11*, *PEX14*, *PEX21*) involved in peroxisome biogenesis and their regulators (*PXA2*, *PIP2*)[25–27]. Tunicamycin treatment induced peroxisome proliferation in *ire1Δ*, *hac1Δ*, and *gcn4Δ* cells, indicating that a functional UPR is not required for peroxisome proliferation during ER stress (Fig. 2a–c). Cells also activate the ER stress surveillance (ERSU) pathway, which prevents the inheritance of damaged ER into daughter cells in response to ER stress[28]. Activation of the ERSU pathway, using phytosphingosine, an early biosynthetic sphingolipid, did not increase the number of peroxisomes (Fig. 2d), Pot1-GFP levels (Fig. 2e), UPR induction (Supplementary Fig. 2a), or HSE-GFP levels (Supplementary Fig. 2b), to the extent seen following tunicamycin treatment. These observations suggest that proteotoxic ER stress causes peroxisome proliferation in a UPR- and ERSU-independent manner (Fig. 2f).

### Misfolded protein stress causes peroxisome proliferation partially via heat shock response activation

Kar2 also functions in the passage of proteins across the Sec61-Sec63 translocon in the ER membrane[29]. By binding to the translocation substrate polypeptide, Kar2 acts as a molecular ratchet to prevent its back-translocation, thereby facilitating its unidirectional movement only towards the ER lumen[30]. We investigated if peroxisome proliferation observed in *kar2-159* cells is due to HSR activation caused by impaired ER-protein translocation (Fig. 3a). We blocked ER protein targeting, without directly interfering with Kar2 or the Sec61-Sec63 translocon, by impairing the GET pathway, which mediates the insertion of Tail-anchored (TA) proteins into the ER[31]. As recently described[32], we observed that peroxisome number increased in the absence of Get3 (Fig. 3b, c). Absence of Get3 also increased HSE-GFP levels at all temperatures examined between 25 °C and 37 °C (Fig. 3d and Supplementary Fig. 3a), indicating the activation of heat shock response by mistargeted TA proteins.

Analogous to the UPR in the ER, the heat shock response pathway is activated in response to the presence of misfolded and aggregated proteins in the cytosol. Signaling through this pathway is primarily mediated by Hsf1, which binds to the Heat Shock Elements (HSEs) in gene promoters and activates transcription of chaperones and components of the Ubiquitin-proteasome system (UPS) involved in protein degradation[33]. We induced the heat shock response pathway by deleting *SSA1*, which encodes the Hsp70 chaperone that binds and represses Hsf1[34]. Under non-stress conditions, Ssa1 and its paralog Ssa2 maintain Hsf1 in an inactive state; however, during stress, unfolded or misfolded proteins titrate the Hsp70s away from Hsf1, thereby allowing transcriptional activation at HSEs. The *ssa1Δ* cells exhibited not

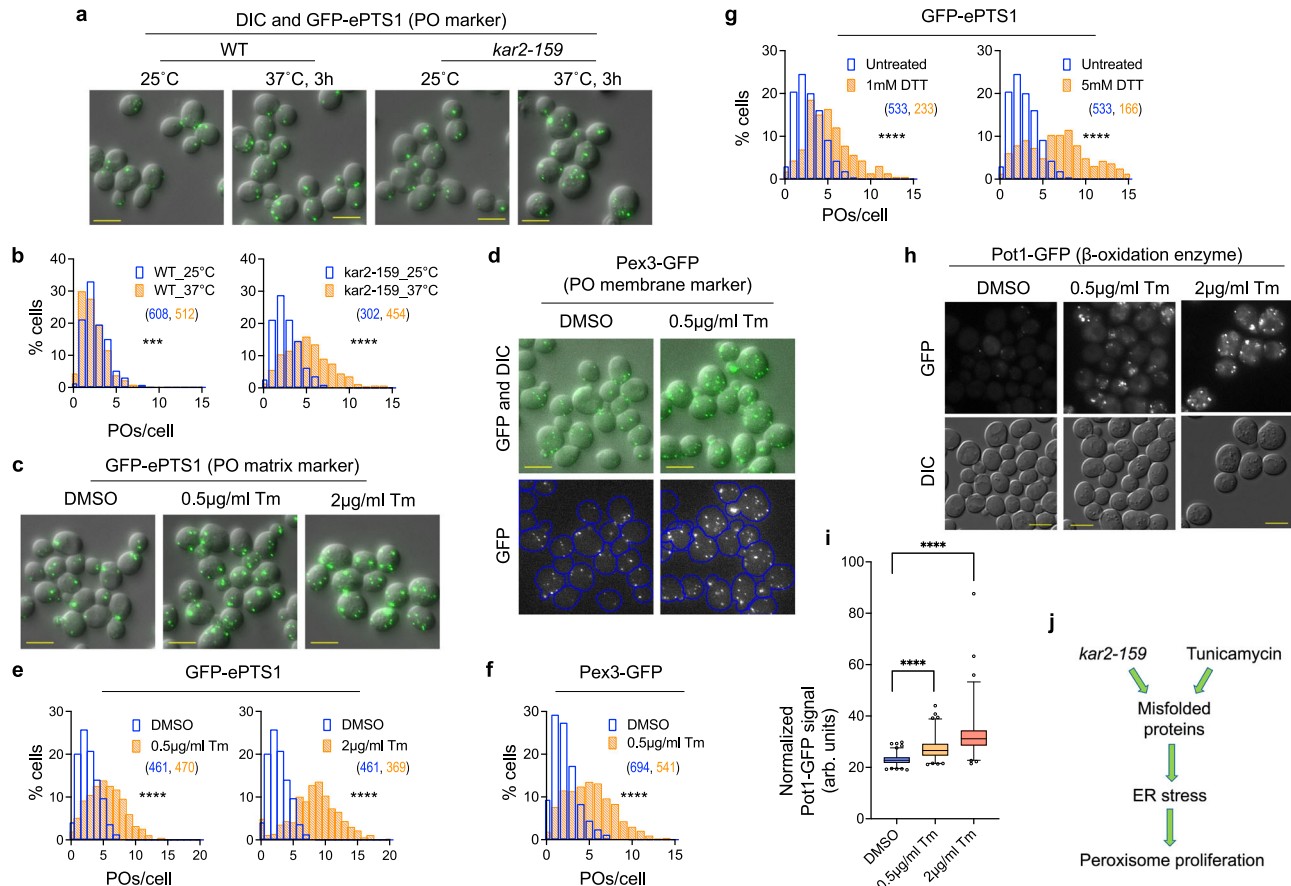

**Fig. 1 | Protein misfolding stress induces peroxisome proliferation.**
**a** Z-projection images showing peroxisomes marked by GFP-ePTS1 in WT and *kar2-159* cells grown at 25 °C and at 3 h after transfer to 37 °C. **b** Histograms showing population distribution of the number of peroxisomes per cell (POs/cell) in WT and *kar2-159* mutant cells at 25 °C and after 3 h of growth at 37 °C [Median POs/cell at 25 °C: WT: 2, *kar2-159*: 2; Median POs/cell at 37 °C: WT: 2, *kar2-159*: 5; number of cells ($N_{cells}$) indicated in parentheses; two-tailed Mann–Whitney test: ***$P < 0.001$; ****$P < 0.0001$]. **c–f** Z-projection images (**c**, **d**) and histograms (**e**, **f**) showing the number of peroxisomes at 5 h after treatment with DMSO or tunicamycin (Tm, 0.5 μg/mL or 2.0 μg/mL as shown). Peroxisomes were visualized using either a peroxisomal matrix marker, GFP-ePTS1 (**c**, **e**), or a peroxisomal membrane protein, Pex3-GFP (**d**, **f**) [$N_{cells}$ indicated in parentheses; two-tailed Mann–Whitney test: ****$P < 0.0001$]. **g** Histograms showing peroxisomes per cell at 5 h after treatment

with 1 mM or 5 mM DTT [$N_{cells}$ indicated in parentheses; two-tailed Mann–Whitney test: ****$P < 0.0001$]. **h**, **i** Single Z-slice images (**h**) and quantification (**i**) showing Pot1-GFP (peroxisomal thiolase) levels per cell after treatment with tunicamycin [$N_{cells}$: DMSO: 510, 0.5 μg/ml Tm: 454, 2 μg/ml Tm: 340; bounds of the box: inter-quartile range (IQR), center line: median, whiskers: 1–99 percentile, points: top and bottom 1 percentile; two-tailed Mann–Whitney test: ****$P < 0.0001$]. **j** Schematic depicting that cells experiencing ER stress, which could be caused either by loss of Kar2 function or treatment with tunicamycin, manifest peroxisome proliferation phenotypes, such as increased organelle number and increased level of peroxisomal enzymes. In this and subsequent figures, the number of cells for each histogram is indicated in parentheses and depicted in matching font color as the histograms. Scale bar in all images: 5 μm.

only elevated HSE-GFP levels (Fig. 3e and Supplementary Fig. 3b), but also a marked increase in peroxisome number compared to WT (Fig. 3b, c). The median number of peroxisomes in *ssa1Δ* was 7–8, versus 3 in WT, with ~70% of *ssa1Δ* cells containing >5 peroxisomes, compared to ~5% in WT. Loss of Ssa1 caused an increase in the number of peroxisomes, comparable to that of tunicamycin-treated wild-type cells (Fig. 3f, g). Surprisingly, tunicamycin further increased the number of peroxisomes in *ssa1Δ* cells, with the median rising to 9–12 after tunicamycin treatment, compared to 6–7 in DMSO-treated cells (Fig. 3f, g). These results suggest that tunicamycin and Hsf1 activation may act synergistically in stimulating peroxisome proliferation.

We compared the effect of tunicamycin on *ssa1Δ* vs. WT cells using a generalized linear modeling (GLM) approach (see "Methods"). Briefly, peroxisome count data was fit to a hurdle model[35], using a GLM model formula containing terms that capture the effects of experimental batch, strain and treatment, as well as an interaction term that captures the strain-specific response to treatment. We then examined the value of the GLM coefficient on the interaction term to compare mutant vs. WT responses to tunicamycin. The statistical significance of

these comparisons was estimated using likelihood ratio tests that compared GLM results using the full model formula to results from a reduced model where the interaction term is omitted. While simply comparing the median number of peroxisomes between strains in an experimental batch provides a cursory glimpse into the relative effects of tunicamycin on those strains, our GLM approach accounts for several important features of our data to increase the statistical power, resolution, and completeness of our comparisons. First, our experimental batches consisted of 1–4 mutant strains and a WT control. Using GLM with a model formula that includes a term for experimental batch effects allowed comparison of peroxisomes between a mutant strain of interest and WT, using data pooled from all experimental batches, thereby maximizing statistical power. Second, some mutant strains had basal cellular peroxisome distributions that were noticeably different from WT. Our GLM approach ensured that these basal differences were accounted for when comparing mutant and WT tunicamycin responses. Third, whereas comparisons of median peroxisome counts have a limited resolution of 0.5, our GLM approach allowed us to quantify tunicamycin responses with a continuous

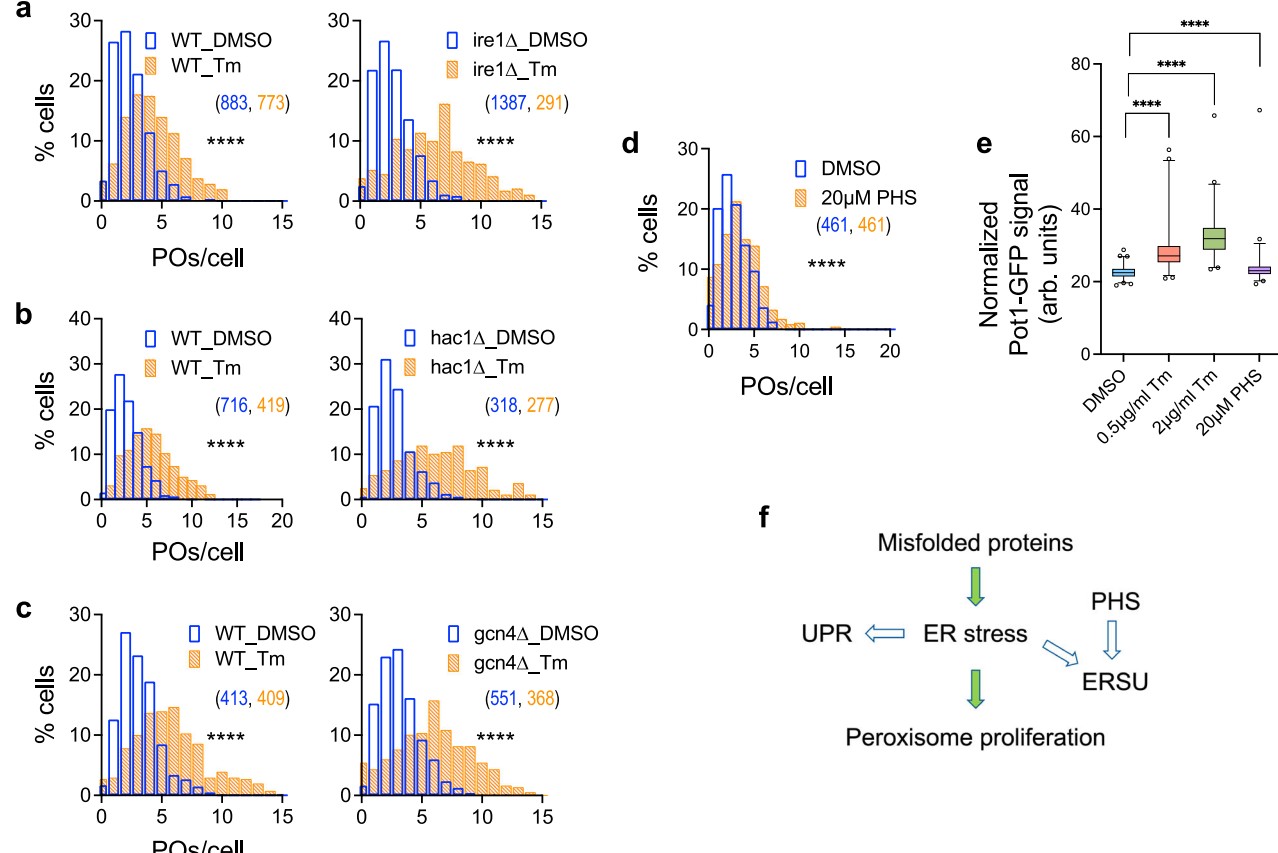

**Fig. 2 | ER stress-induced peroxisome proliferation occurs independent of the UPR and ERSU pathways. a–c** Histograms showing tunicamycin-induced peroxisome proliferation in the UPR pathway mutants, such as *ire1Δ*, *hac1Δ* and *gcn4Δ*, relative to WT cells [$N_{cells}$ indicated in parentheses; two-tailed Mann–Whitney test: ****$P < 0.0001$]. **d, e** Histogram of number of peroxisomes per cell (**d**) and box plot for normalized Pot1-GFP levels (**e**) at 5 h after treatment with phytosphingosine (PHS) or tunicamycin [Median POs/cell: DMSO: 2, 20 μM PHS: 3 for (**d**); $N_{cells}$: DMSO: 327, 0.5 μg/ml Tm: 264, 2 μg/ml Tm: 227, 20 μM PHS: 228 for (**e**); for (**d**) $N_{cells}$

indicated in parentheses; two-tailed Mann–Whitney test: ****$P < 0.0001$; for (**e**) bounds of the box: IQR, center line: median, whiskers: 1–99 percentile, points: top and bottom 1 percentile; two-tailed Mann–Whitney test: ****$P < 0.0001$].
**f** Flowchart showing the potential signaling routes that induce peroxisome proliferation during stress. The green arrows show the routes which we experimentally verified to be operational for signaling, whereas the clear arrows show the path which we verified to either not be required, viz. UPR, or not be sufficient, viz PHS treatment, to induce peroxisome proliferation.

variable. Our analysis revealed that the extent of peroxisome proliferation induced by tunicamycin in *ssa1Δ* cells was less than that in WT cells, suggesting that the loss of Ssa1 and the addition of tunicamycin promote peroxisome proliferation through partially overlapping mechanisms (Fig. 4g).

**Peroxisome proliferation in response to ER stress is partially mediated by TOR1 inactivation**

The adaptive response to ER stress occurs via activation of SNF1-, HOG- and RTG-signaling pathways and inactivation of TOR1 and PKA[27,36–40]; however, among these, the specific pathways that mediate the signaling response to trigger peroxisome proliferation in response to tunicamycin remain unclear (Fig. 4a). We systematically abrogated signaling through each pathway and investigated the effect of tunicamycin on peroxisome proliferation in mutants compared to WT using our GLM analysis (Fig. 4). Given published findings on the involvement of these pathways in yeast peroxisome proliferation and regulation of *PEX* gene expression, we were surprised to discover that blocking the retrograde signaling (RTG) (Fig. 4b, g)[9,11], SNF1[4,8,41] (Fig. 4c, g), or HOG pathways[7] (Fig. 4c, g) did not significantly impair peroxisome proliferation upon tunicamycin treatment. Similarly, the loss of both Msn2 and Msn4 (Fig. 4d, g), which mediate the transcriptional response upon activation of the HOG pathway, as well as the heat shock response pathway[7], also did not prevent tunicamycin-induced peroxisome proliferation. Furthermore, we also found that cells lacking Mig1, the

peroxisome biogenesis repressor that gets phosphorylated and subsequently inactivated by active Snf1, or Mig2, which shares overlapping functions with Mig1[42–44], showed a comparable number of peroxisomes to WT, after both DMSO as well as tunicamycin treatments (Fig. 4e, g). Consistent with our findings that peroxisome proliferation in ER stress is independent of Snf1, we observed that Snf1 was not activated after tunicamycin treatment under our assay conditions; however, Hog1 was activated after 4 h (Supplementary Fig. 4).

Response to ER stress also involves the inhibition of PKA[45] and TOR1[38] (Supplementary Fig. 4). We tested the role of PKA by using a strain overexpressing TPK1 (TPK1oe) using a multicopy plasmid[46], which constitutively activates PKA, relative to a control strain carrying an empty vector. Our GLM analysis indicated that the relative increase in peroxisome number in response to tunicamycin treatment was reduced in TPK1oe cells compared to that in cells carrying the empty vector (Fig. 4f, g). This result suggests that high levels of PKA activity attenuate the tunicamycin-induced peroxisome proliferation.

Finally, we tested the effect of TOR1 inhibition on peroxisomes. After rapamycin treatment, while we observed a mild increase in the number of peroxisomes in *S. cerevisiae* (Fig. 5a, b), the effect in the methylotrophic yeast, *K. phaffii*, was significantly greater, exceeding the increase seen with tunicamycin addition, as indicated by a longer right tail of the distribution in rapamycin-treated, compared to tunicamycin-treated cells (Fig. 5c–e). We also noted that the effects of tunicamycin and rapamycin in *K. phaffii* required twenty-fold higher

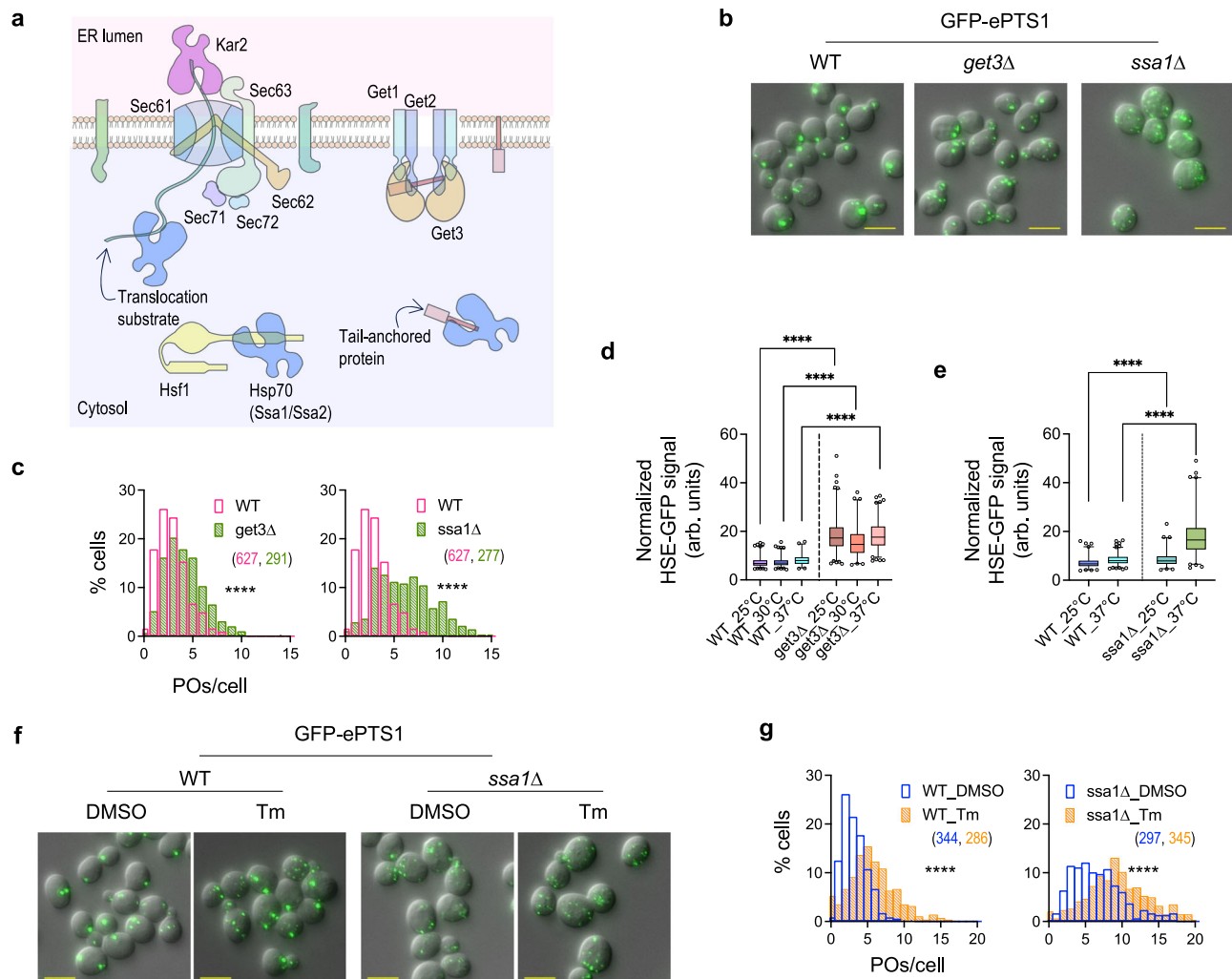

**Fig. 3 | Peroxisome proliferation during ER stress is partially driven via HSR activation.** **a** Cartoon showing molecular players involved in the crosstalk between protein homeostasis in the ER and cytosol. Proteins destined for the ER are trans-located across the ER membrane or inserted via the Sec61 or Get pathways. Kar2 on the luminal side and Hsp70s including Ssa1 in the cytoplasm prevent the translo-cation substrates from misfolding. Hsp70s also keep tail-anchored proteins from misfolding in the cytoplasm before their transfer to Get3 for membrane insertion. Hsp70s additionally are bound to Hsf1 and keep it inactive; however, increased burden of unfolded proteins in the cytoplasm can titrate Hsp70 away from Hsf1, thereby activating the latter. Protein translocation defects in *kar2-159* at 37 °C, or in *get3Δ* can potentially cause the buildup of ER proteins in the cytoplasm, thereby activating the heat shock response. **b, c** Z-projection images (**b**) and histograms (**c**) showing the number of peroxisomes per cell in WT, *get3Δ* and *ssa1Δ* [for (**c**), N$_{cells}$ indicated in parentheses; two-tailed Mann–Whitney test: ****$P < 0.0001$].
**d, e** Induction of HSR visualized using HSE-GFP levels per cell in *get3Δ* at 25 °C or at 3 h after transfer to 30 °C or 37 °C (**d**) or *ssa1Δ* at 25 °C or at 3 h after transfer to 37 °C (**e**) [**d** N$_{cells}$ for WT: 25 °C: 504, 30 °C: 540, 37 °C: 270; N$_{cells}$ for *get3Δ*: 25 °C: 579, 30 °C: 316, 37 °C: 537; **e** N$_{cells}$ for WT: 25 °C: 416, 37 °C: 620; N$_{cells}$ for *ssa1Δ*: 25 °C: 344, 37 °C: 491); **d, e** bounds of the box: IQR, center line: median, whiskers: 1–99 percentile, points: top and bottom 1 percentile; two-tailed Mann–Whitney test: ****$P < 0.0001$]. **f, g** Z-projection images (**f**) and histograms (**g**) showing the number of peroxisomes per cell in *ssa1Δ* compared to WT after treatment with DMSO or 0.5 µg/ml tunicamycin for 5 h [**g** N$_{cells}$ indicated in parentheses; two-tailed Mann–Whitney test: ****$P < 0.0001$]. Scale bar in all images: 5 µm.

(10 µg/ml) drug concentration relative to *S. cerevisiae*, presumably due to robust efflux systems or reduced uptake efficiency in the former, and that tunicamycin treatment reduced viability to ~63% (Supple-mentary Fig. 5a). However, we confirmed that the relatively lower efficacy of rapamycin in *S. cerevisiae* was not due to sub-optimal drug concentration because increasing the rapamycin concentration (5 µg/ml) did not increase peroxisome proliferation (Fig. 5a, b). Overall, the above findings indicate that among the many signaling pathways that are activated by ER stress, activation of the heat shock response pathway and inactivation of TOR1 and PKA are the primary mechan-isms causing peroxisome proliferation.

We tested the significance of peroxisomes in adaptation to ER stress by measuring the survival of WT and *pex3Δ* cells in rich media containing 2% or 0.5% dextrose (where respiration is more prevalent) after overnight (~14 h) treatment with a low dose of 0.5 µg/ml

tunicamycin followed by a high dose of 20 µg/ml tunicamycin for 6 h (Fig. 5f and Supplementary Fig. 5b). Corroborating similar findings in *S. cerevisiae*[9], *K. phaffii* cells lacking Pex3, which is essential for peroxi-some biogenesis[2], consistently showed <50% survival in the tunicamycin-viability assay, which was reduced to <10% under reduced glucose availability compared to WT cells, underscoring the sig-nificance of peroxisome biogenesis in the adaptation.

To investigate the conservation of the peroxisome response to ER stress, we treated primary human fibroblasts with tunicamycin, rapa-mycin, or 4-phenyl butyric acid (4-PBA), a known inducer of peroxi-somes, for 12 h. Cells were fixed, stained for PMP70, a peroxisomal ATP-binding cassette transporter that facilitates the transport of acyl-CoA esters across the peroxisomal membrane, and imaged (Fig. 6a). All three treatments increased peroxisomal density compared to DMSO control as determined by counting the number of PMP70 puncta per unit cell

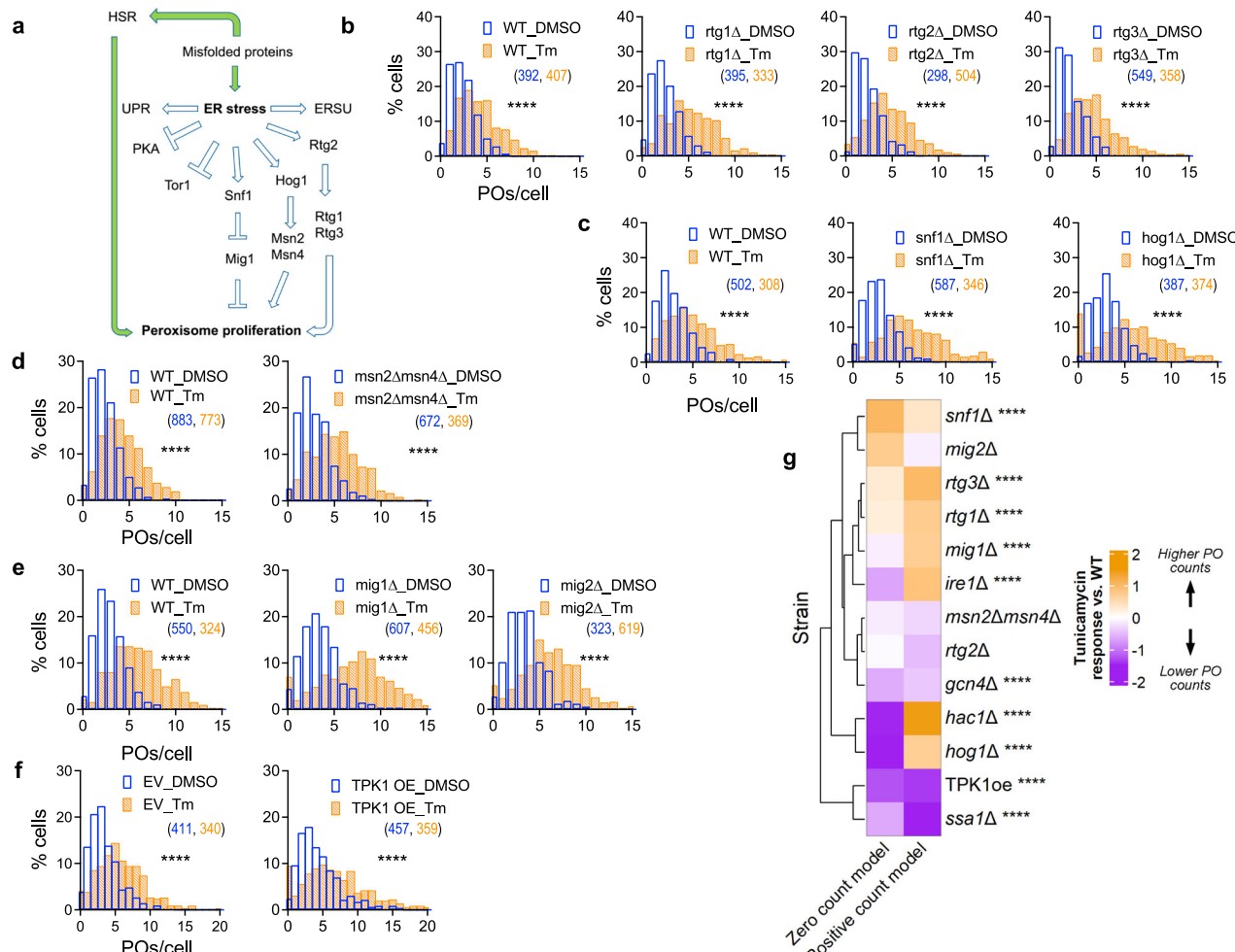

**Fig. 4 | Peroxisome proliferation in response to misfolded protein stress occurs despite individual inactivation of Snf1, Hog1, and Rtg pathways.** **a** Flowchart showing the pathways that get activated or inhibited in response to ER stress. **b**–**e** Histograms showing the number of peroxisomes per cell after 5 h treatment with tunicamycin or DMSO in mutants blocked in signaling through various pathways implicated in peroxisome biogenesis. The effect of blocking the RTG pathway on peroxisome number (**b**) was examined using deletion mutants of the pathway's transcriptional activators, Rtg1 and Rtg3, as well as their upstream regulator, Rtg2. The necessity of SNF1 pathway for peroxisome proliferation upon tunicamycin treatment was tested using *snf1Δ* (**c**), whereas the requirement of the HOG pathway was tested using *hog1Δ* (**c**) and a double deletion mutant of the pathway's transcriptional activators, Msn2 and Msn4 (**d**). WT in (**d**) same as Fig. 2a.

The role of Mig1 and Mig2, transcriptional repressors of peroxisome biogenesis genes, in tunicamycin-driven peroxisome proliferation was tested using their respective deletion mutants (**e**). **f** Constitutive activation of Protein Kinase A (PKA) by overexpressing Tpk1 (TPK1oe) was carried out using a multicopy plasmid containing the *TPK1* ORF expressed from its endogenous promoter [**b**–**f** $N_{cells}$ indicated in parentheses; two-tailed Mann–Whitney test: ****$P < 0.0001$]. **g** Heatmap of GLM analysis showing the mutants with reduced response to tunicamycin in purple and those with increased response in orange, compared to WT. The zero count and positive count columns show response coefficients from their respective components of the hurdle model used to fit the mutant's peroxisomal count data [**** indicates FDR-adjusted $P$ values (Benjamini–Hochberg method) <0.0001 from likelihood ratio tests].

volume (Fig. 6b). These findings demonstrate that the peroxisome proliferation response to ER stress is conserved in human cells.

Peroxisome contribution to cellular adaptation during ER stress was analyzed by comparing the survival of human primary fibroblasts derived from a Zellweger Spectrum Disorder patient carrying homozygous PEX16 R176* mutations (hereafter referred to as *pex16KO*) to WT[47]. Fibroblasts were first pretreated for 12 h with primary doses of tunicamycin (range 0 to 20 μg/ml), then challenged with a secondary tunicamycin treatment (20 μg/ml) or DMSO for 12 h and then assayed for viability. The sensitivity to primary tunicamycin treatment alone was more blunted in *pex16KO* cells compared to WT (Fig. 6c, "DMSO" panel), possibly the result of higher metabolic activity in the latter. When primary treatment was followed with secondary tunicamycin challenge, the prior exposure to tunicamycin conferred a protective effect against subsequent stress in both cell lines, but this benefit was diminished in *pex16KO* cells (Fig. 6c, "Tm" panel). A GLM analysis of this data revealed a significant interaction between genotype, primary

tunicamycin dose, and secondary tunicamycin challenge (likelihood ratio test $P$ value = 0.004), indicating that prior exposure to tunicamycin confers a greater adaptation to secondary challenge in WT versus *pex16KO* cells. These findings suggest that peroxisome biogenesis supports adaptive resilience during ER stress, although the residual benefit observed in *pex16KO* cells suggests the possibility of additional compensatory pathways.

## ER-stress-driven peroxisome proliferation occurs by de novo biogenesis as well as growth and division

We systematically analyzed the contributions of peroxisome biogenesis, via growth and division and de novo pathways, and peroxisome loss by pexophagy or bulk autophagy, to ER-stress mediated peroxisome proliferation (Fig. 7a). We observed that the loss of Atg1, the kinase essential for initiating autophagy, or the pexophagy receptor, Atg36[48], did not increase basal peroxisome numbers (Fig. 7b, c) suggesting that blocking autophagy or pexophagy does

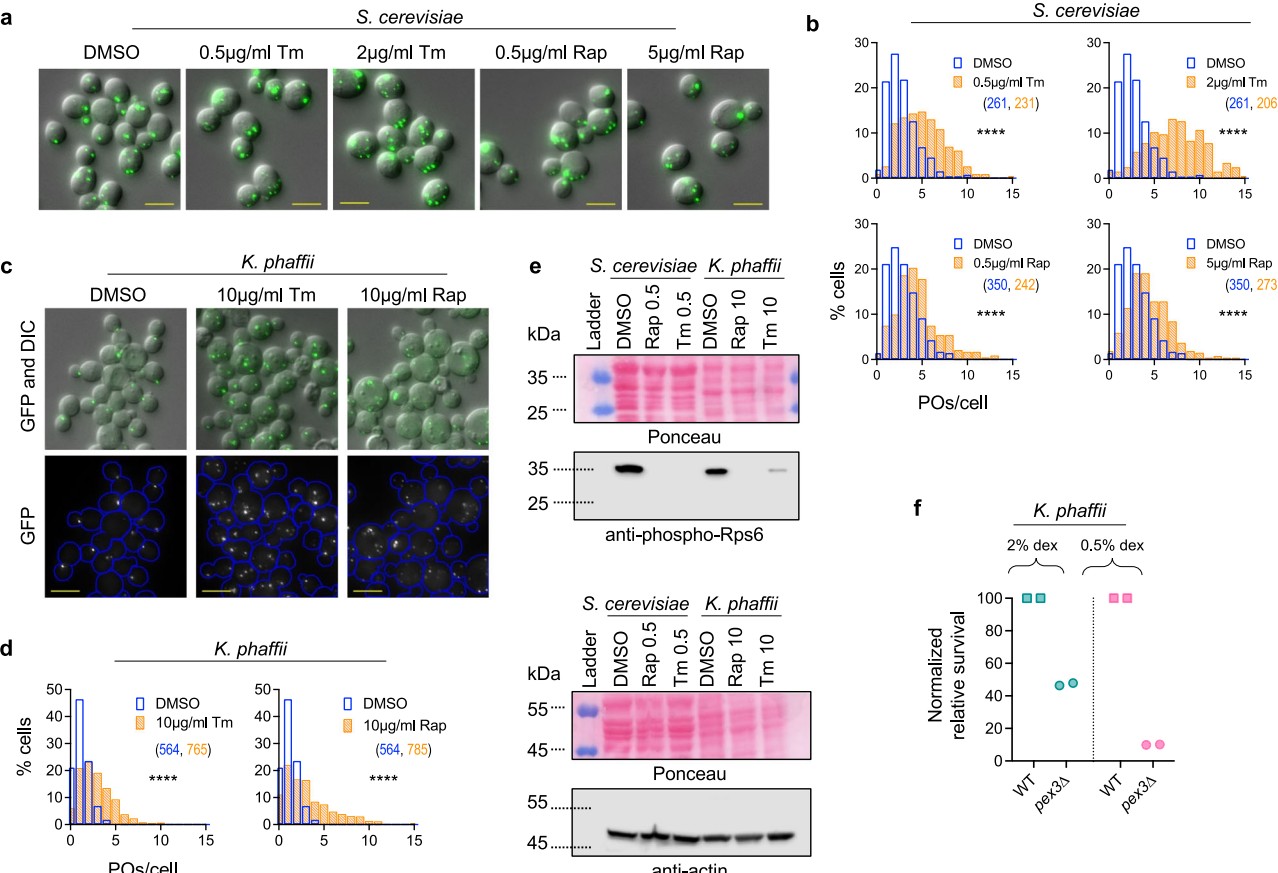

**Fig. 5 | Misfolded protein stress causes peroxisome proliferation in yeast via inactivation of Tor1. a**, **b** Z-projection images (**a**) and histograms (**b**) showing the number of peroxisomes per cell in *S. cerevisiae* after Tor1 inactivation by treatment with 0.5 μg/ml or 5 μg/ml rapamycin treatment. Treatments with DMSO and tunicamycin (0.5 μg/ml and 2 μg/ml) performed for comparison [**b** N$_{cells}$ indicated in parentheses; two-tailed Mann–Whitney test: ****$P < 0.0001$]. **c**, **d** Z-projection images (**c**) and histograms (**d**) showing the number of peroxisomes per cell in the methylotrophic yeast, *K. phaffii*, after treatments with 10 μg/ml rapamycin or 10 μg/ml tunicamycin for 5 h [For (d), N$_{cells}$ indicated in parentheses; two-tailed Mann–Whitney test: ****$P < 0.0001$]. **e** Western blot to examine TOR1 inactivation by testing for Rps6 phosphorylation after treatment of *S. cerevisiae* and *K. phaffii* cells

with DMSO, rapamycin or tunicamycin for 4 h. Ponceau and Actin used as loading and sample processing controls, respectively. The numbers next to the drug names in the labels denote the concentration (μg/ml) of the respective drug used for the treatment. ($n = 1$ for Rap, $n = 3$ for Tm; uncropped blots shown in Source Data, replicates shown in Supplementary Data 2). **f** Quantification of survival of WT and *pex3Δ K. phaffii* cells after overnight treatment with 0.5 μg/ml tunicamycin, followed by treatment with 20 μg/ml tunicamycin for 6 h. Control treatments were performed with DMSO, and colony counts obtained for tunicamycin treatments were divided by those for DMSO treatments to obtain relative survival. Relative survival was normalized with WT to obtain the normalized relative survival for *pex3Δ*. Data from two independent experiments are shown ($n = 2$). Scale bar in all images: 5 μm.

not mimic ER-stress driven peroxisome proliferation. Moreover, *atg36Δ* cells exhibited higher numbers of peroxisomes than WT after tunicamycin treatment (Fig. 7b, e), suggesting that some peroxisomes are likely turned over during ER stress. These findings indicate that peroxisome proliferation during ER stress is not driven by impairments in autophagy or pexophagy.

We next tested whether the increase in peroxisome number during ER stress depended on peroxisome division or on their de novo synthesis. Vps1 is the primary mediator of peroxisome fission in cells grown in the presence of glucose[49], and its deletion resulted in only a partial reduction in tunicamycin-driven peroxisome proliferation compared to WT cells (Fig. 7d, e). We also quantified this tunicamycin response in *dnm1Δ* single and *vps1Δ dnm1Δ* double mutants. While the loss of *DNM1* alone had an insignificant effect, the absence of both *DNM1* and *VPS1* significantly, but still only partially, impaired peroxisome proliferation after tunicamycin treatment (Fig. 7d, e), suggesting additional peroxisome formation occurs by the de novo pathway. In DMSO-treated cells, the median number of peroxisomes was 1 in both *vps1Δ* and *vps1Δ dnm1Δ* mutants, while tunicamycin-treated cells had a median number of 2 peroxisomes per cell. This suggests that tunicamycin treatment led to the de novo formation of at least one additional

peroxisome. In comparison, WT cells had a median number of 3 peroxisomes in DMSO treatment, and 4.5 peroxisomes after tunicamycin treatment. Thus, of the ~1.5 additional peroxisomes formed in WT in response to tunicamycin, at least one could be attributed to de novo biogenesis. Overall, our results demonstrate that de novo biogenesis accounts for approximately two-thirds of the peroxisome formation in response to ER stress.

Further evidence supporting the role of de novo biogenesis in increased peroxisome production during ER stress comes from analyzing the proportion of cells containing peroxisomes in the inheritance mutants *inp2Δ* and *inp1Δ* (Fig. 7f, g). In the absence of Inp2, most daughter cells fail to inherit peroxisomes from mother cells and therefore rely exclusively on de novo biogenesis to produce their first peroxisome[50,51]. Conversely, most mother cells fail to retain peroxisomes in the absence of Inp1[52], resulting in the total transfer of peroxisomes to their daughter cells. We observed ~45% of *inp2Δ* cells lacked peroxisomes under DMSO-treated conditions; however, tunicamycin treatment reduced this proportion to ~22% (Fig. 7g), strongly suggesting that ER stress stimulated de novo biogenesis. Corroborating this observation, we found ~25% of *inp1Δ* cells lacked peroxisomes after tunicamycin treatment, compared to ~40% under DMSO

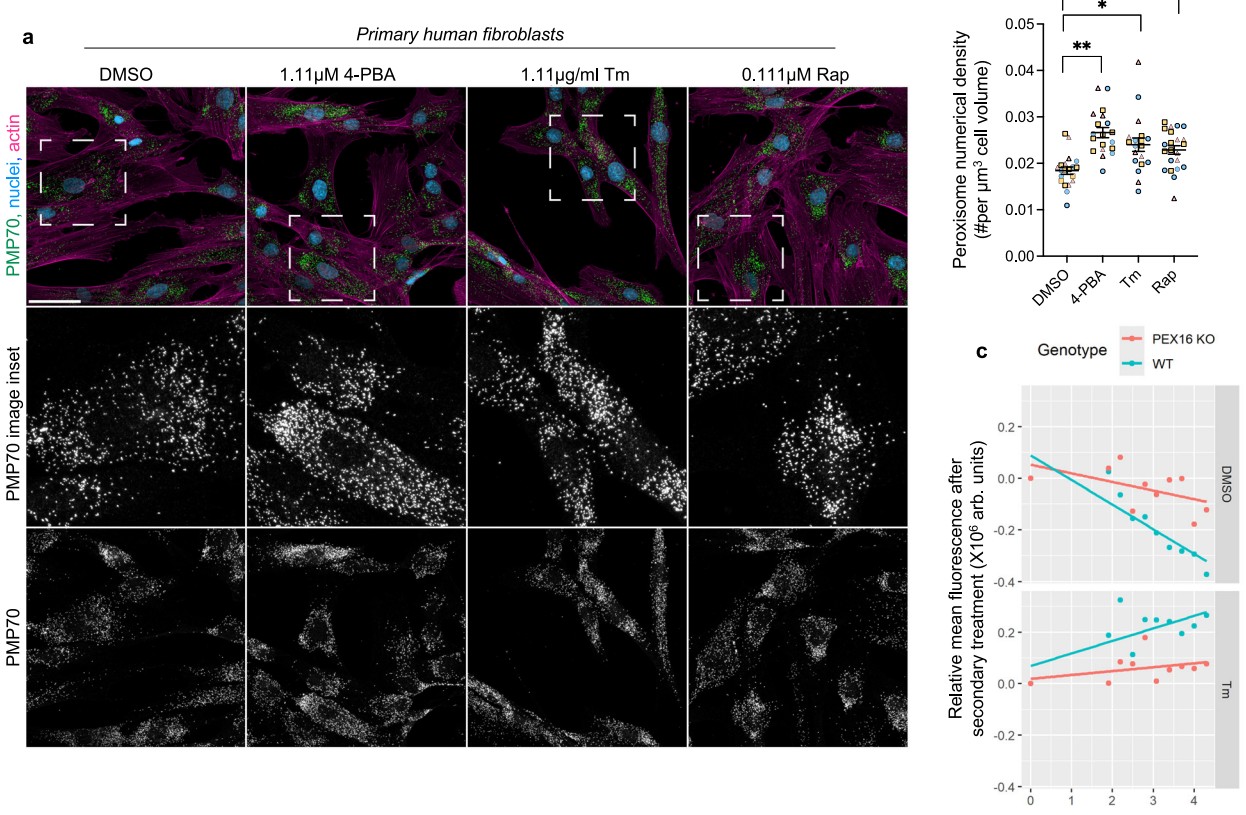

**Fig. 6 | Proteotoxic stress and TOR inactivation cause peroxisome proliferation in primary human fibroblasts. a** Z-projection images showing peroxisome proliferation in primary human fibroblasts after treatment with DMSO, 1.11 μM 4-PBA, 1.11 μg/ml tunicamycin, 0.111 μM rapamycin. Nuclei (blue), peroxisomes (green) and actin (magenta) visualized using DAPI, PMP70-AF488 and Phalloidin-AF750 staining, respectively. Scale bar: 50 μm. **b** Numerical density of peroxisomes per μm³ of cell volume was computed for 5–8 fields of view after independent treatments with either DMSO or the indicated drugs ($n = 3$). Quantification from each field of view is represented by either a circle, triangle or square; each shape represents an independent treatment [For each condition, the mean and standard error of the combined pool of replicates is shown in the plot: DMSO: $0.018 \pm 0.0008$; 4-PBA: $0.027 \pm 0.0011$; Tm: $0.024 \pm 0.0014$; Rap: $0.023 \pm 0.00092$;

two-tailed Nested $t$ test: 4-PBA vs DMSO: $P = 0.0037$; df = 4; Tm vs DMSO: $P = 0.0261$, df = 4; Rap vs DMSO: $P = 0.0223$, df = 4; $*P < 0.05$; $**P < 0.01$]. **c** Quantification of viability assayed by CellTiter-Blue fluorescence assay in WT and *pex16KO* human fibroblasts after a primary, 12 h-long treatment with a range of tunicamycin concentrations, followed by a secondary challenge treatment with 20 μg/ml tunicamycin or DMSO for 12 h. Primary treatments performed with: 0 ng/ml, 80 ng/ml, 155 ng/ml, 310 ng/ml, 625 ng/ml, 1.2 μg/ml, 2.5 μg/ml, 5.0 μg/ml, 10 μg/ml, and 20 μg/ml tunicamycin. Points show mean viability values across at least 24 replicates at the 12 h post-challenge timepoint that were corrected to remove spatial artifacts on experimental plates using a GLM model ("Methods"). Lines are trendlines fit to these points.

treatment conditions (Fig. 7g). Since de novo peroxisome biogenesis in *S. cerevisiae* is slow, requiring ~4–5 h[50,53], and our analyses were performed within this same timeframe, most of the new peroxisomes observed in response to ER stress are likely due to de novo biogenesis, with the remaining, comprising about one-third, likely derived from the growth and division.

**Impact of lipid homeostasis on the regulation on peroxisome proliferation during ER stress**
Besides misfolded proteins, lipid imbalance also causes ER stress[54]. Moreover, the major function of peroxisomes is to carry out β-oxidation of fatty acids. Therefore, we tested the effects of lipid bilayer stress (LBS) on peroxisomes by either growing cells in media lacking inositol or upon impairing phosphatidylcholine (PC) synthesis by deleting *CHO2* or *OPI3*. While growth of yeast in media lacking inositol induced UPR (Supplementary Fig. 7a) and had a mild effect on HSE-GFP level (Supplementary Fig. 7b), there was no significant increase in the number of peroxisomes (Fig. 8a) and only a slight increase of Pot1-GFP levels (Fig. 8b). However, in comparison to these observations, cells lacking Opi3 or Cho2 displayed more peroxisomes compared to WT (Fig. 8c). Notably, *cho2Δ* and *opi3Δ* trigger the HSR[55] suggesting that

lipid bilayer stress can induce peroxisome proliferation if it activates HSR.

Lipid droplets play important roles in energy metabolism and lipid homeostasis by serving as lipid storage sites in cells. Moreover, not only are lipid droplets and peroxisomes generated from a common site in the ER membrane, but the two organelles also maintain crosstalk through contact sites[2]. Furthermore, ER stress stimulates lipid droplet formation[56], as well as microlipophagy[57], and one possibility is that lipid droplets could serve as the source of fatty acids for β-oxidation by peroxisomes. We examined how peroxisome proliferation in response to ER stress is coordinated with lipid droplet homeostasis. We found that upon impairing lipid droplet formation by using the quadruple deletion (ΔQD) strain, *are1Δ are2Δ lro1Δ dga1Δ*[58], peroxisome number was comparable to WT (Fig. 8d, e). The absence of lipid droplets also did not abrogate the tunicamycin-driven increase in peroxisome number (Figs. 4g and 8e) indicating peroxisome proliferation occurs independent of lipid droplet biogenesis.

## Discussion
While peroxisome induction in response to nutritional cues is well studied, fewer studies have examined the effects of

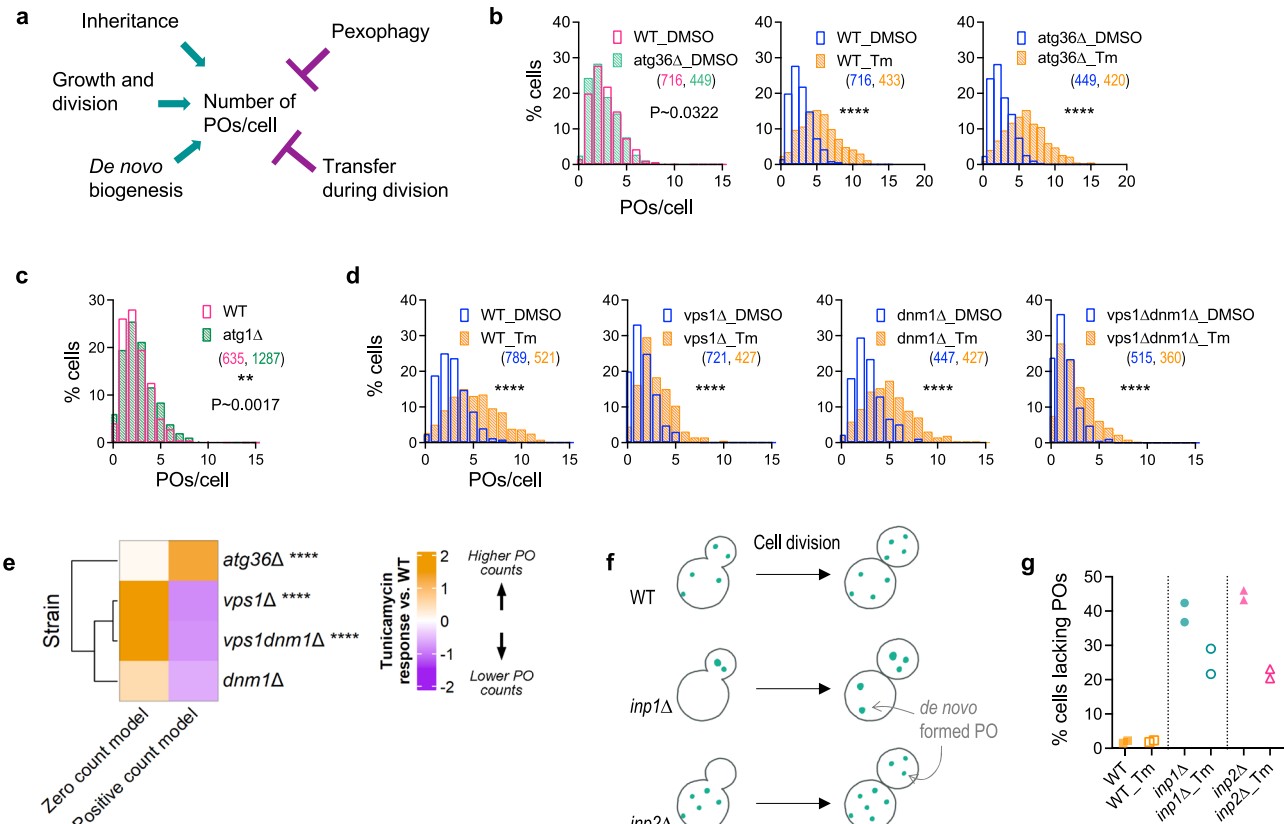

**Fig. 7 | Increased peroxisome number during ER stress occurs by growth and division as well as de novo biogenesis. a** Cartoon showing that the peroxisome number in cells is regulated by a balance between pathways that increase peroxisomes, such as growth and division, de novo biogenesis from the ER and acquisition of peroxisomes due to inheritance, and those that reduce peroxisomes, such as pexophagy or transmission to daughter cells during cell division. **b–d** Histograms showing the number of peroxisomes per cell in (**b**) WT or the pexophagy mutant, *atg36Δ*, after treatment with DMSO or 0.5 μg/ml tunicamycin for 5 h, (**c**) the autophagy mutant, *atg1Δ*, in comparison to WT (Median number of peroxisomes: WT: 2, *atg1Δ*: 2) and (**d**) after 5 h treatment with DMSO or tunicamycin in peroxisome division mutants, *vps1Δ*, *dnm1Δ* and *vps1Δ dnm1Δ* compared to WT [**b–d** N_cells indicated in parentheses; two-tailed Mann–Whitney test; comparison considered significantly different if *P* < 0.05; **P* < 0.01; *****P* < 0.0001; for WT_DMSO vs *atg36Δ*_DMSO in (**b**), *P* = 0.0322; for WT vs *atg1Δ* in (**c**), *P* = 0.0017]. **e** GLM analyses for the comparison of relative effects of tunicamycin treatment on the number of peroxisomes in mutants versus WT [Likelihood ratio test; FDR-adjusted P values using Benjamini–Hochberg method: *****P* < 0.0001]. **f** Cartoon illustrating peroxisome inheritance defects in *inp1Δ* and *inp2Δ* mutants. **g** Quantification of proportion of cells without peroxisomes in the inheritance mutants 5 h after two independent treatments with DMSO or 0.5 μg/mL tunicamycin (*n* = 2).

environmental stress on peroxisomes, especially at the mechanistic level[9–11]. Our study reveals that ER stress induces peroxisome proliferation in *S. cerevisiae*, as well as in *K. phaffii* and human fibroblasts, which represent 250 M and 1 billion years of divergence from *S. cerevisiae*, respectively[59].

Peroxisomes have been considered to be dispensable for growth in glucose since yeast mutants impaired in peroxisome biogenesis are viable and do not exhibit major growth defects. Moreover, enzymes for fatty acid β-oxidation are typically repressed in cells metabolizing glucose. However, our study demonstrates that even during growth in glucose media, upon onset of ER stress, peroxisomes become crucial for survival and the expression of a key β-oxidation enzyme, Pot1, is induced. Additionally, induction of other peroxisomal enzymes required for redox balance, carbon metabolism and amino acid biosynthesis, has also been reported[60]. Thus, ER stress not only induces peroxisomes, but reprograms them for cellular adaptation.

The onset of ER stress triggers several signaling pathways to enable survival via adaptive mechanisms, including peroxisome proliferation (Fig. 8f). Among these, the activation of UPR or ER stress surveillance pathways is not involved in inducing peroxisome proliferation. Instead, our results point to the mistargeting of ER proteins, as seen in *get3Δ* cells, and/or the subsequent accumulation of misfolded proteins in the cytosol, as observed in *ssa1Δ* cells, as primary triggers of peroxisome proliferation. Our findings reveal a mechanism linking protein quality control and cytosolic proteotoxic stress to peroxisome biogenesis. Although the RTG pathway has been previously shown to increase peroxisome numbers during salt stress and Pex11-GFP levels after tunicamycin treatment, and the HOG pathway regulates fatty acid metabolism during salt stress and oleate metabolism, our data reveal that both pathways are dispensable for peroxisome proliferation during ER stress. The Snf1 pathway and its downstream effectors, such as Adr1, Oaf1, and Cat8, regulate peroxisome biogenesis and induction of β-oxidation enzymes in yeast metabolizing oleate[20,25]. However, several lines of evidence indicated that ER-stress-driven peroxisome proliferation is independent of Snf1. Firstly, contrary to a previous report[39], we did not observe Snf1 activation even after 4 h of tunicamycin treatment under our assay conditions. Moreover, loss of Snf1 did not prevent peroxisome proliferation after tunicamycin addition. Furthermore, loss of Mig1, the transcriptional repressor, which is inactivated upon Snf1 activation, or Mig2, the transcriptional repressor with overlapping targets as Mig1 but under the control of Snf3, did not mimic tunicamycin-driven peroxisome proliferation or further increase the effects of tunicamycin on peroxisome number.

Our study has also identified the role of TOR, HSR and PKA pathways in the regulation of stress-induced peroxisomes. Our data indicate that rapamycin treatment and thereby TOR inactivation,

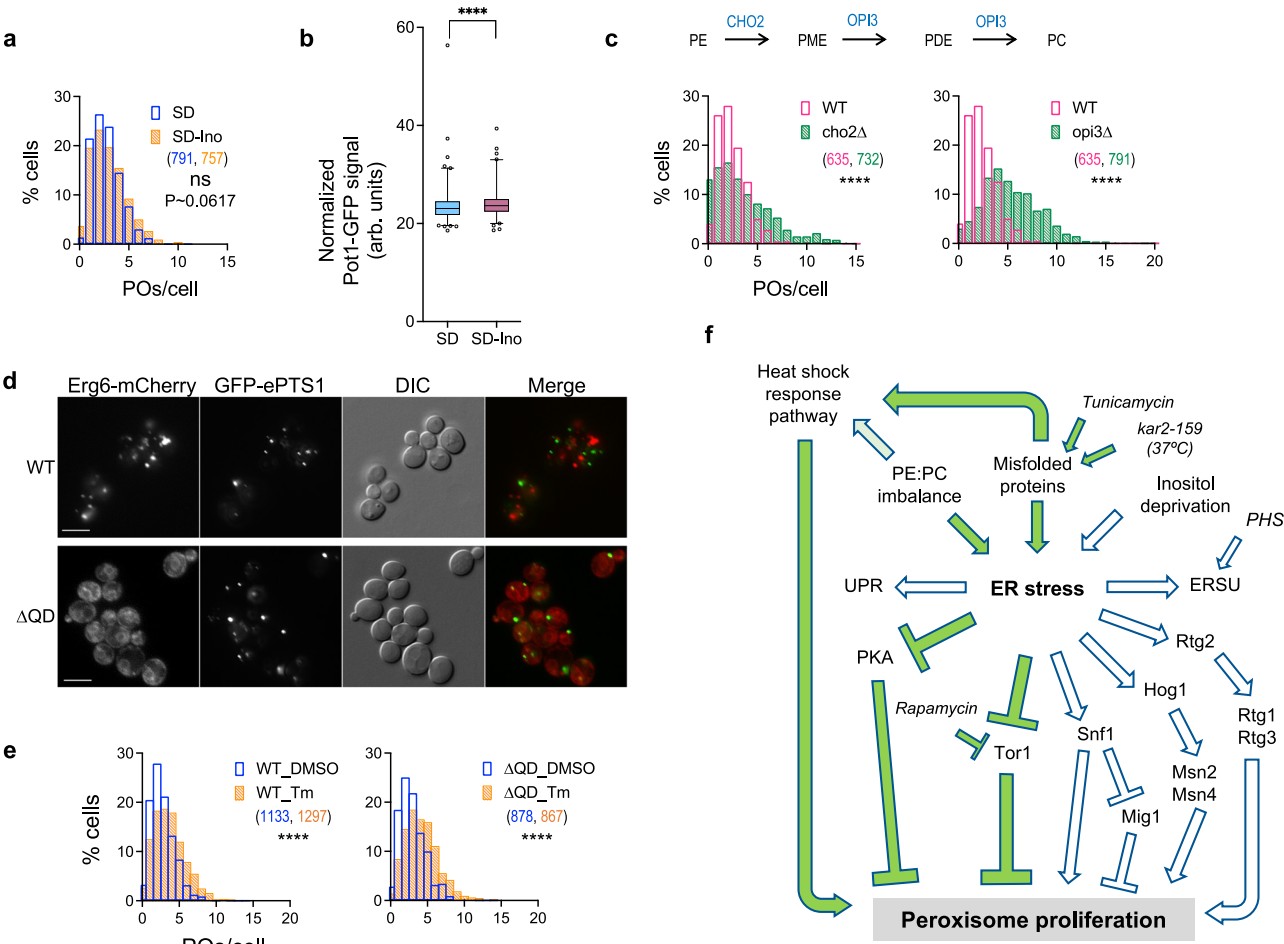

**Fig. 8 | Effects of lipid homeostasis impairments on peroxisome proliferation.**
**a**, **b** Histogram of peroxisomes per cell (**a**) and box plot showing normalized Pot1-GFP levels (**b**) in WT cells grown in SD media versus in the absence of inositol [Median number of peroxisomes: SD: 3, SD-Ino: 3; $N_{cells}$ indicated in parentheses; two-tailed Mann–Whitney test; comparison considered significantly different if $P < 0.05$; 0.0617; **b** $N_{cells}$: SD: 523, SD-Ino: 405]. **c** Pathway for PE to PC conversion (top) and histograms (bottom) showing the number of peroxisomes per cell in mutants impaired in PC synthesis, $cho2\Delta$ and $opi3\Delta$. WT same as Fig. 7c [Median POs/cell: WT: 2, $cho2\Delta$: 3, $opi3\Delta$:5; $N_{cells}$ indicated in parentheses; two-tailed Mann–Whitney test: ****$P < 0.0001$]. **d** Single Z-slice images showing peroxisomes (GFP-ePTS1) and the localization of Erg6-mcherry, the lipid droplet marker, in WT and a quadruple mutant ($\Delta QD$, $are1\Delta$ $are2\Delta$ $lro1\Delta$ $dga1\Delta$) impaired in lipid droplet

formation. Scale bar: 5 μm. **e** Histograms showing the number of peroxisomes per cell in WT and $\Delta QD$ mutant, after treatment with DMSO or 0.5 μg/ml tunicamycin for 5 h [$N_{cells}$ indicated in parentheses; two-tailed Mann–Whitney test: ****$P < 0.0001$]. GLM analysis indicated that the QD mutant's response to Tm was not significantly different than WT (Likelihood ratio test $P$ value = 0.5039).
**f** Flowchart showing that the accumulation of misfolded proteins can lead to peroxisome proliferation via multiple routes including inactivation of TOR1 and PKA pathways, or activation of heat shock response (HSR). While proteotoxic ER stress also leads to activation of UPR-, SNF1-, HOG-, RTG and ERSU-, peroxisome proliferation under these conditions occurs independent of these pathways. In addition, among the pathways causing lipid bilayer stress, PC deficiency also leads to peroxisome proliferation, probably via HSR activation.

either partially in *S. cerevisiae* and mammalian cells, or considerably in *K. phaffii*, mimics the effects of tunicamycin on peroxisomes. This agrees with a previous report of TOR inhibition during ER stress, as corroborated also by our study. ER stress also inhibits PKA, and our data show that PKA activation using TPK1 overexpression partially impairs tunicamycin-driven induction of peroxisomes. Notably, Hsf1 activation can inhibit TOR signaling[61] and overexpression of Hsf1 can activate transcriptional targets of Oaf1, Pip2, and Msn2/4[62]. Hsf1 in turn has been shown to be under negative control of PKA[63]. However, the interplay between Tor1 and PKA pathways is more complex with the two kinases depicting mutually positive, as well as negative, interactions[64].

Among the two pathways of peroxisome formation, yeast cells metabolizing glucose primarily synthesize new peroxisomes by growth and division. However, upon transition to growth in oleate, peroxisomes undergo proliferation with dynamin Dnm1 playing an important role in peroxisome fission. Consistent with this, our data indicate that stress-induced peroxisome proliferation is partially

impaired in cells lacking Vps1, but is independent of Dnm1, suggesting a mechanism distinct from growth in oleate. Furthermore, by comparing the extent of ER-stress-driven peroxisome proliferation in mutants blocked in peroxisome division ($vps1\Delta$ and $vps1\Delta$ $dnm1\Delta$) versus WT, we estimate that de novo biogenesis accounts for roughly two-thirds of peroxisome biogenesis in response to ER stress, with growth and division playing a supportive role. Additional evidence for tunicamycin-induced de novo peroxisome formation comes from $inp1\Delta$ and $inp2\Delta$ cells, which have peroxisomes only in daughter and mother cells[51], and new peroxisomes in mother and daughter cells in these respective mutants can only have arisen de novo[65]. In contrast, salt-induced stress in yeast induces peroxisome by upregulating division of the organelle[11].

Our study also demonstrates that while peroxisome proliferation may be intimately tied to lipid homeostasis, impairing lipid balance by blocking lipid droplet formation did not alter the number of peroxisomes in untreated cells or block peroxisome proliferation after tunicamycin treatment. Lipid stress or UPR activation using inositol

deprivation also does not cause peroxisome proliferation. Notably, while abrogation of PC synthesis, using *cho2Δ* or *opi3Δ* mutants, does cause peroxisome proliferation, its effects could be via HSR activation. Moreover, PC deficiency also leads to the destabilization and degradation of certain ER proteins, including Sbh1, the β-subunit of the Sec61 translocon[66], suggesting that its effects on peroxisome proliferation could be driven via impaired translocation of membrane proteins as seen in *get3Δ* cells.

Overall, this study extends the influence of the ER and the cytosol on peroxisome homeostasis, further emphasizing the interconnectivity and coordination between compartments in mounting an appropriate response to stress and maintaining cellular homeostasis. While ER stress induces peroxisome proliferation, peroxisomal import stress also induces the ER stress response and the integrated stress response (ISR) pathway in flies and human cells[67]. These data and the results presented here establish the framework of an important, conserved, feedback loop between ER and peroxisome stress. In addition, our study may inform pharmacological interventions that can induce peroxisomes in human cells which will be exciting to pursue further in developing therapeutics for PBDs.

## Methods

### Strains and plasmids
Yeast strains were generated using lithium acetate-PEG-mediated transformation and are listed in Supplementary Data 1. Strains expressing $P_{TDH3}$-GFP-ePTS1-$T_{PGK1}$ (i.e., GFP-ePTS1), 4×UPRE-$P_{CYC1}$-GFP-$T_{ACT1}$ (i.e., UPRE-GFP) or 4×HSE-$P_{CYC1}$-GFP-$T_{ACT1}$ (i.e., HSE-GFP) were generated by transforming linear constructs obtained by double digesting the plasmids pNS31, pNS32 or pLT1, respectively, with PciI and NotI. All three constructs were targeted after the STOP codon of the *UBC9* ORF. Except where noted, parental knock-out strains were obtained from the MATα deletion library (Invitrogen)[68]. *SNF1, INP2,* and *DNM1* genes were knocked out by transforming the corresponding deletion constructs, which in turn, were generated by fusing hphMX6 or Zeo cassettes on either side with ~600-1000 bp sequences upstream and downstream of the respective ORFs. Appropriate replacement of these ORFs by the selection marker was confirmed by primers shown in Supplementary Data 2. 4×UPRE-$P_{CYC1}$-GFP-$T_{ACT1}$ was amplified from pRH1209[69]. For generating strains overexpressing *TPK1* (TPK1oe), plasmid pPHY2056[46] was transformed in yeast; the corresponding control strain was made by transforming the empty vector (EV), pRS426 in yeast.

### Yeast growth, treatments, microscopy, and western blotting
Except when noted, *S. cerevisiae* log phase cells grown in synthetic defined [SD; 6.7 g/L Yeast nitrogen base (YNB without amino acids, BD/Difco; 291920) + 0.79 g CSM (Sunrise Science Products; 1001-100) + 2% dextrose] at 30 °C and shaken at 250 RPM were used. For experiments with HSE-GFP and *kar2-159*, cultures were grown at 25 °C and 250 RPM. RTG pathway mutants are glutamate auxotrophs, and hence those mutants and their corresponding WT were grown in SD + 0.02% glutamic acid. Cultures were resuspended in fresh media at an OD ~0.05–0.1 prior to performing temperature shifts (to 30 °C or 37 °C for 3 h as indicated in figure legends) or treatments with tunicamycin (Sigma, T7765; 1 mg/ml stock in DMSO) or rapamycin (Sigma, R0395; 1 mg/ml stock in DMSO) or phytosphingosine (Sigma, Cas No. 554-62-1; 10 mM stock in DMSO) or DMSO or DTT (Roche, 10708984001; 1 M stock in $H_2O$). TPK1 overexpression experiments were performed in SD-Ura media to maintain selection for the plasmids. For all the experiments with *K. phaffii*, cell growth and treatments were done in YPD media (10 g/L Yeast extract + 20 g/L Bactopeptone + 2% Dextrose). SD-Ino media was prepared using YNB without inositol (US Biological Life Sciences; Y2030-01). The effect of tunicamycin on cell viability was measured by quantifying the percentage of cells stained by propidium iodide (PI). For this, yeast cultures were treated with 4 μg/ml PI (Sigma, P4170; 1 mg/ml stock in DMSO) for 20 min at room temperature, followed by 1× wash and resuspension in media for subsequent imaging.

Cells were imaged using a Zeiss Plan Apochromat 63×/1.4 Oil DIC objective mounted on an Axioskop 2 mot plus microscope (Zeiss) equipped with Axio Cam HRm camera and HBO 100 mercury lamp. For 3D imaging to count the number of peroxisomes per cell, Z stacks consisting of 8 slices (for *S. cerevisiae*) or 6 slices (for *K. phaffii*) were acquired with a Z spacing of 1 μm. For UPRE-GFP, HSE-GFP, Pot1-GFP and PI staining only a single slice at the focal plane was imaged. Exposure times were kept identical across all experiments for each fluorescent marker, whereas DIC images were acquired with auto exposure.

Western blotting was performed from cell extracts generated after TCA-precipitation using the following antibodies: Phospho-S6 Ribosomal Protein (Ser235/236) (D57.2.2E) XP Rabbit mAb, Cell Signaling Technology (#4858, 1:2000); Phospho-AMPKα (Thr172) (40H9) Rabbit mAb, Cell Signaling Technology (#2535, 1:2000); Phospho-p38 MAPK (Thr180/Tyr182) (D3F9) XP Rabbit mAb, Cell Signaling Technology (#4511, 1:2000); Phospho-PKA Substrate (RRXS*/T*) (100G7E) Rabbit mAb, Cell Signaling Technology (#9624, 1:2000); Anti-ScActin Rabbit pAb[20], Gift from Michael Yaffe (1:5000); Goat Anti-Rabbit IgG (H + L)-HRP Conjugate, Bio-RAD; (#1721019, 1:5000).

### Human fibroblast cell culture and microscopy
Human primary fibroblasts were cultured at 37 °C with 5% $CO_2$ in high-glucose Dulbecco's Modified Eagle Medium (DMEM; high glucose with sodium bicarbonate and non-essential amino acids; Gibco) supplemented with 15% (v/v) heat-inactivated fetal bovine serum (FBS; VWR). Primary human fibroblasts were purchased from Coriell Institute (Cat. No. GM06231 and GM00969). All experiments were conducted using fibroblasts with fewer than ten passages. For imaging, fibroblasts were seeded in glass-bottom 96-well plates (Cellvis) and treated with the indicated concentrations of tunicamycin (Sigma), rapamycin (Sigma), or 4-phenyl butyric acid (4-PBA) (Sigma). After a 12 h incubation, cells were fixed with formaldehyde and stained with NucBlue Stain mixed with ProLong Glass Antifade Mountant (Thermo Fisher Scientific, P36985), PMP70-AF488 [anti-PMP70 antibody, Abcam (EPR5614; 1:1000); donkey anti-Rabbit IgG (H + L)-Alexa Fluor 488, Thermo Fisher Scientific (A-21206, 1:2000)], and Phalloidin-AF750 (Thermo Fisher Scientific, A30105, 165 nM working conc.). Each treatment condition included three biological replicates. For imaging, 6–8 3D confocal z-stacks per well were acquired using a Zeiss 980 confocal microscope equipped with a 63×/1.2 NA water immersion objective lens and NIR detector. Each z-stack had dimensions of 4096 × 4096 × 31 voxels, corresponding to a physical volume of approximately 210 × 210 × 6 μm³.

### Image processing and quantitative analysis of human fibroblasts
To quantify peroxisome abundance in individual human fibroblasts, we developed a custom 3D image processing pipeline implemented in Python (Clarifi3D), optimized for high-resolution confocal z-stacks and using NVIDIA A100 GPUs on the Seattle Children's Research Institute High-Performance Compute Cluster, Sasquatch. Raw images (4096 × 4096 × 31 voxels; 0.0509 × 0.0509 × 0.2 μm voxel size) were loaded and preprocessed using GPU-accelerated normalization and filtering routines. Cell and nucleus segmentation were performed using pre-trained 3D U-Net models applied to the phalloidin and DAPI channels, respectively, followed by seed-based Delaunay watershed segmentation. Peroxisomes were segmented from PMP70-AF488 images using a classical spot detection pipeline comprising 3D Laplacian-of-Gaussian morphological filtering, global Otsu thresholding, and seeded 3D Delaunay watershed segmentation. For each field-of-view, the numerical peroxisome density was calculated as the number of segmented peroxisomes divided by the total cytoplasmic

volume of fully segmented cells. Results from six to eight stacks per well were averaged across three biological replicates per condition.

## Viability assay in human fibroblasts

Human primary dermal fibroblasts from a healthy donor (GM00969; wild-type, WT) and a ZSD-patient cell line for PEX16 (GM06231; *pex16KO*) line were cultured in DMEM (high glucose, with sodium bicarbonate and non-essential amino acids; Gibco) supplemented with 15% FBS. Cells were maintained at 37 °C in a humidified atmosphere containing 5% $CO_2$. For drug treatments, cells were seeded at 2000 cells per well in black-wall, clear-bottom 384-well plates (Costar) in 40 μL media per well and allowed to adhere overnight. Drug treatments were performed in a randomized matrix format using tunicamycin (dissolved in DMSO) in a 1:2 serial dilution. Tunicamycin was titrated to a top concentration of 20 μg/mL. The viability assay was conducted on two independent 384-well plates, with 12 biological replicates per concentration on each plate. After 12 h of drug exposure, half of the wells received a secondary tunicamycin challenge at 20 μg/mL to induce acute ER stress, while the other half received a DMSO mock treatment. Viability was assessed 12 h later using CellTiter-Blue (Promega) per the manufacturer's instructions. CellTiter-Blue fluorescence was quantified using a Spectromax I3 automated plate imager (Molecular Devices). Raw fluorescence intensity was extracted from each well.

## Data analyses, statistics, and reproducibility

Figure images for depicting the number of peroxisomes in cells were made by generating maximum intensity Z projections for the peroxisome channel (GFP) and average intensity Z projection for cell outline (DIC) channel. Figure images for depicting UPRE-GFP, Pot1-GFP, and HSE-GFP were made with single-Z slice images. Identical brightness and contrast settings were applied across all images within a figure panel to ensure fair comparisons, using FIJI/ImageJ software. Scale bars in all figures represent 5 μm, except for the human fibroblasts, where a 50 μm scale bar was used. Quantification for the number of peroxisomes per cell (for GFP-ePTS1 and Pex3-GFP) as well as for the intensity of GFP signal per cell (for Pot1-GFP, UPRE-GFP and HSE-GFP) was performed using the perox-per-cell software[23] using a minimum peroxisome area threshold of 1 pixel and the following values for the software's peroxisome detection sensitivity: 0.0064 for GFP-ePTS1, Pot1-GFP, 4xUPRE-GFP, and 4xHSE-GFP; 0.003 for Pex3-GFP. For quantification of Pot1-GFP, UPRE-GFP and HSE-GFP levels, the values for GFP signal intensities for each cell were divided by 10,000× the value of their respective cell areas and to obtain the "normalized GFP signal" used in the plots. GraphPad Prism (version 10.4.0, Boston, Massachusetts) and R (version 4.4.0) were used to generate the graphs.

No statistical method was used to predetermine sample size. Except for experiments involving *kar2-159* and *hac1*Δ (rationale explained below), no data were excluded from the analyses. The experiments were not randomized except for experiments in Fig. 6c, where for the human fibroblasts grown in 384-well plates, drug treatments were performed in a randomized manner. The investigators were not blinded to allocation during experiments and outcome assessment. All yeast experiments were done twice, in separate

included both repeats, to rule out discrepancies due to batch-specific effects. In the box plots, the bound of the box indicates the interquartile range (IQR), whereas the center line indicates the medians. The whiskers in the box plots represent 1–25 and 75–99 percentiles of the data, whereas the top and bottom 1 percentiles are represented as dots outside these bounds. In the histograms, the $X$ axis was cropped mostly at 15 or sometimes at 20 peroxisomes/cell, since any higher bin values accounted for <1% of the frequencies; however, these higher data points were not excluded from analyses, and all numerical data are included in Source Data/Supplementary Data 2. We observed significant cell death for *kar2-159* ts mutant cells at 37 °C, and *hac1*Δ after tunicamycin treatment, therefore we manually excluded dead cells from the histograms for those experiments. For this, any atypical cell that showed shriveling, the presence of large vacuoles, or obvious damage and abnormal shape as visualized on the DIC channel was marked as dead. Our empirical observations indicated that almost all cells with zero peroxisomes were dead or severely damaged, except in *inp1*Δ and *inp2*Δ, where the lack of peroxisomes can be attributed to inheritance defects. We also did not exclude dead cells from the data for GLM analyses (see below), hence many of the mutants that appear to have significant changes in zero peroxisome counts are those where we observed more death after tunicamycin treatment (viz. *hac1*Δ, *hog1*Δ). Using GraphPad Prism, after confirming that the data did not fit normal distribution using D'Agostino & Pearson, Anderson–Darling, Shapiro–Wilk, and Kolmogorov–Smirnov tests, we used the Two-tailed Mann–Whitney test and computed a $P$ value for each replicate to evaluate statistical significance between control and treatment groups. An unpaired $t$ test with Welch's correction was used to evaluate statistical significance for comparison of peroxisome area and cumulative peroxisome area per cell ($n = 3$ experiments). Statistical significance for peroxisomal numerical density in human fibroblasts was evaluated using a nested $t$ test ($n = 3$ experiments).

## Quantitative analyses of tunicamycin-induced peroxisome counts across mutant strains

We used generalized linear modeling (GLM) in R (version 4.4.0) to model the distribution of peroxisome (PO) counts in cells across treatment conditions and strains. Due to the presence of a high number of cells with zero counts in some strains, we modeled distributions with hurdle models[35] using the "pscl" R package (version 1.5.9), where zero and non-zero (positive) peroxisome counts are captured using separate functions. A binomial distribution was used to model zero counts, and a negative binomial distribution was used to model non-zero counts. We found that, based on the Akaike Information Criterion, hurdle models fit the data better than zero-inflation and simple negative binomial models.

To compare a mutant strain's tunicamycin response to WT, we first combined that strain's peroxisome count data for both the DMSO and tunicamycin treatment conditions with the full set of WT count data collected across all experiments and strains. This included data from both DMSO- and tunicamycin-treated WT cells. We then used GLM to fit the data using a model formula that included strain, treatment, experimental batch, and an interaction term to quantify strain-specific responses to tunicamycin.

$$\text{FULL MODEL: } PO \text{ count} \sim Strain + Treatment + Batch + Strain : Treatment \tag{1}$$

experimental batches and all numerical data from these repeats organized as batches are available in Supplementary Data 2. A batch consisted of a group of 1–4 mutants that were tested together with a common WT control. Two different batches for a mutant thus represented two independent treatments, each with at least 100 cells. While the graphs in the figures depict data from a single repeat/batch, GLM analyses were performed on data pooled across batches and thus

We tested whether the mutant strain's response to tunicamycin was significantly different than WT by performing a likelihood ratio test where GLM results using the following reduced model were compared to those from the full model.

$$\text{REDUCED MODEL: } PO \text{ count} \sim Strain + Treatment + Batch \tag{2}$$

*P* values collected from these tests across strains were then adjusted for the false discovery rate using the Benjamini–Hochberg method[70]. We considered mutant strains with *P* values < 0.05 as those with tunicamycin responses that were significantly different from WT.

To determine whether a mutant strain's response to tunicamycin was stronger or weaker compared to WT, we examined the value of the coefficients for the *Strain:Treatment* interaction term in the mutant strain's full GLM model. Because hurdle models have two modeling components, there are two interaction term coefficients to consider: one from the zero-count model and one from the non-zero count model. For heatmap visualization, the coefficients from a given hurdle modeling component were z-score normalized across the values obtained from all strains (i.e., normalized by column for the heatmaps). For consistency, positive values in these visualizations indicate that the modeling component predicts a stronger tunicamycin response in a mutant strain compared to WT. That is, it predicts a response resulting in higher peroxisome counts. As mentioned earlier, we observed a significantly large proportion of dead cells in *hac1Δ* and *hog1Δ* strains following tunicamycin treatment. The zero-count model interaction term coefficients in the GLM analyses on those strains reflected this, resulting in higher predicted zero-count instances. To ensure that the presence of dead cells did not bias our analyses, we made inferences on the relative effects of tunicamycin-induced peroxisome proliferation for mutants vs. WT based only on the interaction term coefficients of the positive-count model.

### Quantitative analysis of WT and *pex16KO* human fibroblast survival to tunicamycin-induced ER stress

To quantitatively characterize the contribution of peroxisomes to the adaptive response of human fibroblasts to ER stress, we performed a GLM analysis on the viability assay data from WT and *pex16KO* human fibroblast cells described above ("Viability assay in human fibroblasts"). Using the CellTiter-Blue fluorescence readout as an indicator of viability, we used R to fit our data using the following formula.

secondary treatment were then normalized by subtracting the value observed at a primary tunicamycin concentration of 0 ng/mL. Trendlines in plots were computed using linear regression.

### Reporting summary
Further information on research design is available in the Nature Portfolio Reporting Summary linked to this article.

## Data availability
All data generated in this study are provided in the supplementary information/source data files. Source data are provided with this paper.

## Code availability
The code for image processing of yeast cells is available at https://github.com/AitchisonLab/perox-per-cell. The code for image processing of human cells is available at https://github.com/fdmast-2/CLARIFI3D.

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

$$Viability \sim Genotype * PrimaryTmConcentration * SecondaryTreatment + Row + Column + Timepoint \qquad (3)$$

Here, *PrimaryTmConcentration* represents the concentration of tunicamycin used to pretreat cells prior to secondary challenge. The *SecondaryTreatment* term indicates the secondary challenge applied to the cells (DMSO or tunicamycin). The *Row* and *Column* terms were included to account for spatial artifacts on experimental plates. Two post-challenge measurements were performed, one at 12 h and one at 12.5 h, and the *Timepoint* term is included to quantify viability differences between those two timepoints.

To determine whether there was a difference between strains in the way that tunicamycin pretreatment affected viability following secondary challenge, we performed a likelihood ratio test comparing results from the full model formula above to results from a reduced model where the interaction term *Genotype:PrimaryTmConcentration:SecondaryTreatment* was removed. This is the term in the model that captures how the strains respond to the secondary treatment stress with respect to the primary concentration of tunicamycin they received. The value of this term's coefficient indicates the degree to which primary tunicamycin treatment concentration alters WT cells' viability following secondary tunicamycin challenge compared to *pex16KO* cells. For scatterplots used to visualize differences in strain responses to secondary challenge, we used our fitted GLM to correct raw experimental fluorescence values for spatial artifacts on experimental plates, then computed the mean values across replicates. All mean values for a specific combination of genotype and

6. Lawrence, J. W. et al. Differential gene regulation in human versus rodent hepatocytes by peroxisome proliferator-activated receptor (PPAR) alpha. PPAR alpha fails to induce peroxisome proliferation-associated genes in human cells independently of the level of receptor expression. *J. Biol. Chem.* **276**, 31521–31527 (2001).
7. Rajvanshi, P. K., Arya, M. & Rajasekharan, R. The stress-regulatory transcription factors Msn2 and Msn4 regulate fatty acid oxidation in budding yeast. *J. Biol. Chem.* **292**, 18628–18643 (2017).
8. Saleem, R. A. et al. Genome-wide analysis of signaling networks regulating fatty acid-induced gene expression and organelle biogenesis. *J. Cell Biol.* **181**, 281–292 (2008).
9. Hijazi, I., Wang, E., Orozco, M., Pelton, S. & Chang, A. Peroxisomal support of mitochondrial respiratory efficiency promotes ER stress survival. *J. Cell Sci.* https://doi.org/10.1242/jcs.259254 (2022).
10. Hinojosa, L. et al. Impact of heat and drought stress on peroxisome proliferation in quinoa. *Plant J.* **99**, 1144–1158 (2019).
11. Manzanares-Estreder, S., Espi-Bardisa, J., Alarcon, B., Pascual-Ahuir, A. & Proft, M. Multilayered control of peroxisomal activity upon salt stress in *Saccharomyces cerevisiae*. *Mol. Microbiol* **104**, 851–868 (2017).
12. Di Cara, F., Savary, S., Kovacs, W. J., Kim, P. & Rachubinski, R. A. The peroxisome: an up-and-coming organelle in immunometabolism. *Trends Cell Biol.* **33**, 70–86 (2023).
13. Pan, R., Liu, J. & Hu, J. Peroxisomes in plant reproduction and seed-related development. *J. Integr. Plant Biol.* **61**, 784–802 (2019).

14. Berger, J., Dorninger, F., Forss-Petter, S. & Kunze, M. Peroxisomes in brain development and function. *Biochim Biophys. Acta* **1863**, 934–955 (2016).

15. Honsho, M., Okumoto, K., Tamura, S. & Fujiki, Y. Peroxisome biogenesis disorders. *Adv. Exp. Med. Biol.* **1299**, 45–54 (2020).

16. Musa, M. et al. Lack of peroxisomal catalase affects heat shock response in *Caenorhabditis elegans*. *Life Sci. Alliance* https://doi.org/10.26508/lsa.202201737 (2023).

17. Kaur, N., Li, J. & Hu, J. Peroxisomes and photomorphogenesis. *Subcell. Biochem.* **69**, 195–211 (2013).

18. Ayer, A. et al. Distinct redox regulation in sub-cellular compartments in response to various stress conditions in *Saccharomyces cerevisiae*. *PLoS ONE* **8**, e65240 (2013).

19. Fransen, M. & Lismont, C. Redox signaling from and to peroxisomes: Progress, challenges, and prospects. *Antioxid. Redox Signal* **30**, 95–112 (2019).

20. Farre, J. C., Carolino, K., Devanneaux, L. & Subramani, S. OXPHOS deficiencies affect peroxisome proliferation by downregulating genes controlled by the SNF1 signaling pathway. *eLife*. https://doi.org/10.7554/eLife.75143 (2022).

21. Wu, F., de Boer, R. & van der Klei, I. J. Gluing yeast peroxisomes - composition and function of membrane contact sites. *J. Cell Sci.* https://doi.org/10.1242/jcs.259440 (2023).

22. Silva, B. S. C. et al. Maintaining social contacts: the physiological relevance of organelle interactions. *Biochim. Biophys. Acta Mol. Cell Res.* **1867**, 118800 (2020).

23. Neal, M. L. et al. Automated, image-based quantification of peroxisome characteristics with perox-per-cell. *Bioinformatics*. https://doi.org/10.1093/bioinformatics/btae442 (2024).

24. Kimata, Y. et al. Genetic evidence for a role of BiP/Kar2 that regulates Ire1 in response to accumulation of unfolded proteins. *Mol. Biol. Cell* **14**, 2559–2569 (2003).

25. Smith, J. J. et al. Transcriptional responses to fatty acid are coordinated by combinatorial control. *Mol. Syst. Biol.* **3**, 115 (2007).

26. Natarajan, K. et al. Transcriptional profiling shows that Gcn4p is a master regulator of gene expression during amino acid starvation in yeast. *Mol. Cell Biol.* **21**, 4347–4368 (2001).

27. Patil, C. K., Li, H. & Walter, P. Gcn4p and novel upstream activating sequences regulate targets of the unfolded protein response. *PLoS Biol.* **2**, E246 (2004).

28. Pina, F. et al. Sphingolipids activate the endoplasmic reticulum stress surveillance pathway. *J. Cell Biol.* **217**, 495–505 (2018).

29. Young, B. P., Craven, R. A., Reid, P. J., Willer, M. & Stirling, C. J. Sec63p and Kar2p are required for the translocation of SRP-dependent precursors into the yeast endoplasmic reticulum in vivo. *EMBO J.* **20**, 262–271 (2001).

30. Sanders, S. L., Whitfield, K. M., Vogel, J. P., Rose, M. D. & Schekman, R. W. Sec61p and BiP directly facilitate polypeptide translocation into the ER. *Cell* **69**, 353–365 (1992).

31. Schuldiner, M. et al. The GET complex mediates insertion of tail-anchored proteins into the ER membrane. *Cell* **134**, 634–645 (2008).

32. Bittner, E. et al. Proteins that carry dual targeting signals can act as tethers between peroxisomes and partner organelles. *PLoS Biol.* **22**, e3002508 (2024).

33. Hahn, J. S., Hu, Z., Thiele, D. J. & Iyer, V. R. Genome-wide analysis of the biology of stress responses through heat shock transcription factor. *Mol. Cell Biol.* **24**, 5249–5256 (2004).

34. Masser, A. E. et al. Cytoplasmic protein misfolding titrates Hsp70 to activate nuclear Hsf1. *eLife*. https://doi.org/10.7554/eLife.47791 (2019).

35. Cragg, J. G. Some statistical models for limited dependent variables with applications to the demand for durable goods. *Econometrica* **39**, 829–844 (1971).

36. Appenzeller-Herzog, C. & Hall, M. N. Bidirectional crosstalk between endoplasmic reticulum stress and mTOR signaling. *Trends Cell Biol.* **22**, 274–282 (2012).

37. Ahmed, K., Carter, D. E. & Lajoie, P. Hyperactive TORC1 sensitizes yeast cells to endoplasmic reticulum stress by compromising cell wall integrity. *FEBS Lett.* **593**, 1957–1973 (2019).

38. Hijazi, I., Knupp, J. & Chang, A. Retrograde signaling mediates an adaptive survival response to endoplasmic reticulum stress in *Saccharomyces cerevisiae*. *J. Cell Sci.* https://doi.org/10.1242/jcs.241539 (2020).

39. Mizuno, T., Masuda, Y. & Irie, K. The *Saccharomyces cerevisiae* AMPK, Snf1, negatively regulates the Hog1 MAPK pathway in ER stress response. *PLoS Genet.* **11**, e1005491 (2015).

40. Bicknell, A. A., Tourtellotte, J. & Niwa, M. Late phase of the endoplasmic reticulum stress response pathway is regulated by Hog1 MAP kinase. *J. Biol. Chem.* **285**, 17545–17555 (2010).

41. Ratnakumar, S. & Young, E. T. Snf1 dependence of peroxisomal gene expression is mediated by Adr1. *J. Biol. Chem.* **285**, 10703–10714 (2010).

42. Westholm, J. O. et al. Combinatorial control of gene expression by the three yeast repressors Mig1, Mig2 and Mig3. *BMC Genomics* **9**, 601 (2008).

43. Papamichos-Chronakis, M., Gligoris, T. & Tzamarias, D. The Snf1 kinase controls glucose repression in yeast by modulating interactions between the Mig1 repressor and the Cyc8-Tup1 co-repressor. *EMBO Rep.* **5**, 368–372 (2004).

44. Treitel, M. A., Kuchin, S. & Carlson, M. Snf1 protein kinase regulates phosphorylation of the Mig1 repressor in *Saccharomyces cerevisiae*. *Mol. Cell Biol.* **18**, 6273–6280 (1998).

45. Pincus, D., Aranda-Diaz, A., Zuleta, I. A., Walter, P. & El-Samad, H. Delayed Ras/PKA signaling augments the unfolded protein response. *Proc. Natl. Acad. Sci. USA* **111**, 14800–14805 (2014).

46. Budovskaya, Y. V., Stephan, J. S., Reggiori, F., Klionsky, D. J. & Herman, P. K. The Ras/cAMP-dependent protein kinase signaling pathway regulates an early step of the autophagy process in *Saccharomyces cerevisiae*. *J. Biol. Chem.* **279**, 20663–20671 (2004).

47. South, S. T. & Gould, S. J. Peroxisome synthesis in the absence of preexisting peroxisomes. *J. Cell Biol.* **144**, 255–266 (1999).

48. Motley, A. M., Nuttall, J. M. & Hettema, E. H. Atg36: the *Saccharomyces cerevisiae* receptor for pexophagy. *Autophagy* **8**, 1680–1681 (2012).

49. Hoepfner, D., van den Berg, M., Philippsen, P., Tabak, H. F. & Hettema, E. H. A role for Vps1p, actin, and the Myo2p motor in peroxisome abundance and inheritance in *Saccharomyces cerevisiae*. *J. Cell Biol.* **155**, 979–990 (2001).

50. Motley, A. M. & Hettema, E. H. Yeast peroxisomes multiply by growth and division. *J. Cell Biol.* **178**, 399–410 (2007).

51. Fagarasanu, M., Fagarasanu, A. & Rachubinski, R. A. Sharing the wealth: peroxisome inheritance in budding yeast. *Biochim Biophys. Acta* **1763**, 1669–1677 (2006).

52. Fagarasanu, M., Fagarasanu, A., Tam, Y. Y., Aitchison, J. D. & Rachubinski, R. A. Inp1p is a peroxisomal membrane protein required for peroxisome inheritance in *Saccharomyces cerevisiae*. *J. Cell Biol.* **169**, 765–775 (2005).

53. Hoepfner, D., Schildknegt, D., Braakman, I., Philippsen, P. & Tabak, H. F. Contribution of the endoplasmic reticulum to peroxisome formation. *Cell* **122**, 85–95 (2005).

54. Fun, X. H. & Thibault, G. Lipid bilayer stress and proteotoxic stress-induced unfolded protein response deploy divergent transcriptional and non-transcriptional programmes. *Biochim. Biophys. Acta Mol. Cell Biol. Lipids* **1865**, 158449 (2020).

55. Brandman, O. et al. A ribosome-bound quality control complex triggers degradation of nascent peptides and signals translation stress. *Cell* **151**, 1042–1054 (2012).

56. Fei, W., Wang, H., Fu, X., Bielby, C. & Yang, H. Conditions of endoplasmic reticulum stress stimulate lipid droplet formation in *Saccharomyces cerevisiae*. *Biochem J.* **424**, 61–67 (2009).

57. Garcia, E. J. et al. Membrane dynamics and protein targets of lipid droplet microautophagy during ER stress-induced proteostasis in the budding yeast, *Saccharomyces cerevisiae*. *Autophagy* **17**, 2363–2383 (2021).

58. Choudhary, V., El Atab, O., Mizzon, G., Prinz, W. A. & Schneiter, R. Seipin and Nem1 establish discrete ER subdomains to initiate yeast lipid droplet biogenesis. *J. Cell Biol.* https://doi.org/10.1083/jcb.201910177 (2020).

59. Bernauer, L., Radkohl, A., Lehmayer, L. G. K. & Emmerstorfer-Augustin, A. *Komagataella phaffii* as emerging model organism in fundamental research. *Front. Microbiol.* **11**, 607028 (2020).

60. Xie, J., Xiao, C., Pan, Y., Xue, S. & Huang, M. ER stress-induced transcriptional response reveals tolerance genes in yeast. *Biotechnol. J.* **19**, e2400082 (2024).

61. Bandhakavi, S. et al. Hsf1 activation inhibits rapamycin resistance and TOR signaling in yeast revealed by combined proteomic and genetic analysis. *PLoS ONE* **3**, e1598 (2008).

62. Hou, J. et al. Management of the endoplasmic reticulum stress by activation of the heat shock response in yeast. *FEMS Yeast Res.* **14**, 481–494 (2014).

63. Lee, P., Cho, B. R., Joo, H. S. & Hahn, J. S. Yeast Yak1 kinase, a bridge between PKA and stress-responsive transcription factors, Hsf1 and Msn2/Msn4. *Mol. Microbiol.* **70**, 882–895 (2008).

64. Plank, M. Interaction of TOR and PKA signaling in *S. cerevisiae*. *Biomolecules*. https://doi.org/10.3390/biom12020210 (2022).

65. Wroblewska, J. P. & van der Klei, I. J. Peroxisome maintenance depends on de novo peroxisome formation in yeast mutants defective in peroxisome fission and inheritance. *Int. J. Mol. Sci.* https://doi.org/10.3390/ijms20164023 (2019).

66. Shyu, P. Jr. et al. Membrane phospholipid alteration causes chronic ER stress through early degradation of homeostatic ER-resident proteins. *Sci. Rep.* **9**, 8637 (2019).

67. Kim, J. et al. Peroxisomal import stress activates integrated stress response and inhibits ribosome biogenesis. *PNAS Nexus* **3**, 429 (2024).

68. Giaever, G. et al. Functional profiling of the *Saccharomyces cerevisiae* genome. *Nature* **418**, 387–391 (2002).

69. Sato, B. K., Schulz, D., Do, P. H. & Hampton, R. Y. Misfolded membrane proteins are specifically recognized by the transmembrane domain of the Hrd1p ubiquitin ligase. *Mol. Cell* **34**, 212–222 (2009).

70. Benjamini, Y. & Hochberg, Y. Controlling the false discovery rate: A practical and powerful approach to multiple testing. *J. R. Stat. Soc: Series B (Methodol.)* **57**, 289–300 (1995).

## Acknowledgements

This study was funded by grant NIDDK 41737 to S.S. and J.D.A. S.S. is a Tata Chancellor Professor in Molecular Biology. NS is grateful for partial support from the Molecular Biology Cancer Fellowship (UC San Diego). F.D.M. is the recipient of a career development award from Seattle Children's Research Institute. We are thankful to Qihao Liu and Lorraine Pillus (UC San Diego) for constructing the *msn2Δ msn4Δ* strain, and Haley Schultz for the sHS60 strain. We are also grateful to Brenda Andrews and Charlie Boone (Univ. of Toronto), Patrick Lajoie (The Univ. of Western Ontario), Tasuya Maeda (Univ. of Tokyo), Paul Herman (The Ohio State Univ.), Randolph Hampton (UC San Diego), William Prinz (UT Southwestern) and Richard Rachubinski (Univ. of Alberta) for strains and plasmids. We gratefully acknowledge the support of Andrew Longenecker and the PBD Project for providing assistance with human fibroblasts used in this study. The authors acknowledge Research Scientific Computing at Seattle Children's Research Institute for providing HPC resources that have contributed to the research results reported within this paper.

## Author contributions

N.S. and S.S. co-conceived the study, interpreted data, and co-wrote the first draft. N.S. generated strains and plasmids, designed and performed experiments, and analyzed the data. M.N. performed GLM analyses. J.-C.F. discovered the role of Tor1 inhibition on peroxisome number, performed experiments on lipid homeostasis, viability as well as all experiments in *K. phaffii*. F.D.M. and L.R.M. performed human cell experiments. R.S. performed experiments on *pex* mutants. L.T. and N.S. performed experiments on UPRE-GFP and HSE-GFP. T.S. assisted in strain generation. J.D.A. and S.S. supervised the study and obtained funding. N.S., S.S., M.L.N., J.-C.F., F.D.M., and J.D.A. edited the manuscript.

## Competing interests

The authors declare no competing interests.
