## [Transparent Peer Review file · Nature Communications]

TOR and heat shock response pathways regulate peroxisome biogenesis during proteotoxic stress

Corresponding Author: Dr Suresh Subramani

Version 0:

Reviewer comments:

Reviewer #1

(Remarks to the Author)

This study demonstrates that misfolded protein stress in the ER induces peroxisome proliferation in yeast. The process is partially dependent on the TOR1 and heat shock proteins. There is an extensive analysis of whether stress induced peroxisome proliferation requires various kinases and transcription factors. These are fascinating findings since there was no previous evidence that peroxisome proliferation could be driven by unfolded protein in the ER. There are several ways the study could be improved.

1. Does peroxisome proliferation following ER stress help cells survive? For example, are pex mutants more sensitive to ER stress or do they recover from tunicamycin or DTT treatment more poorly than wt cells? Do mammalian cells that lack peroxisomes or that fail to proliferate them following ER stress have defects recovering from ER stress? More investigation of the function of stress induced peroxisome proliferation is necessary for this study to be appropriate for Nat Comm.
2. The results in Fig. 5g should be quantified. Please show higher mag images without the actin staining so it is easier to see the peroxisomes. Determining whether proliferation occurs in multiple cell types would strengthen the study.
3. It would be good to know whether the amounts of tunicamycin and DTT used to cause ER stress are lethal to the cells. If they are, what fraction of the cells are viable after 5 hours of treatment? Please also show peroxisomes proliferation over time.
4. Only one mutant is used to determine whether lipid stress induces peroxisome proliferation. Other mutant that significantly alter ER lipid composition could be tried such as cho1 and cho2.
5. The study shows that blocking pexophagy does not affect ER expansion, but what about blocking bulk autophagy?
6. The discussion is very long and includes a lot of material that seems better suited for a review or that could be said more concisely. The authors may want to consider shortening it.

Reviewer #2

(Remarks to the Author)

In this manuscript, Shukla et al. describe experiments arguing that protein misfolding stress induces peroxisome proliferation. They exposed cells to various pharmacologic/chemical, genetic, and environmental perturbations and measured the number of peroxisomes in each case. While most of their experiments were done in budding yeast, they extended some of these studies to methylotrophic yeast cells and human fibroblasts. From all these experiments, the conclusion was that inputs leading to protein misfolding/stress would increase peroxisome copy number. The information likely involves the heat shock response pathways and an antagonistic role of the TOR pathway. Overall, the manuscript is interesting, and the experiments are done well. However, after reading the manuscript, the reader is left with the sense that the story is very descriptive and does not move the field significantly, given the well-established role of other perturbations that induce peroxisome biogenesis, such as oxidative stress.

- The authors measure peroxisome copy number as a proxy for peroxisome biogenesis. Presumably, changes in the total peroxisomal compartment (i.e., the sum of all individual peroxisomes in each cell) more accurately reflect the relationship between stress and peroxisomes and the metabolic roles of these organelles. But they focus exclusively on copy numbers. Is the size of individual peroxisomes similar in all the conditions they test? Are there cases where there might be an increase in copy number (more but smaller peroxisomes) that does not significantly change the total peroxisomal volume per cell?
- Is the response they observe separate from the oleate-induced peroxisome biogenesis? Their results with *dnm1* deletants argue that they are. But are the two responses additive? What happens if they add oleate on top of the various stressors they test?
- Related to their role in fatty acid metabolism, what happens to lipid droplets in these stresses, and how peroxisome biogenesis is affected in stressed and unstressed cells if lipid droplet homeostasis is perturbed genetically? Likewise, what happens if fatty acid synthesis is affected (e.g., by blocking fatty acid synthase with drugs, such as cerulenin, or by relevant genetic perturbations)?
- Following up on the above, contact sites between peroxisomes and lipid droplets are known to be dynamic. How are these sites affected by the various stresses tested?
- Their summary Figure 7 is a typical 'hairball' signaling figure, where everything regulates everything. Is it possible to replace it with a simpler, more meaningful one?

Reviewer #3

(Remarks to the Author)

Kar2 activation has also been reported with a defect in *Sec59* (James et al. 2019, 10.1038/s41598-019-51054-7). Looking at *Sec61*-*Sec63* selectively provides only a partial picture. The authors need to study the role of *Sec59*.

Figure 1: An important control is missing. The authors need to study the distribution pattern of GFP alone in WT and mutant cells.

How does the developed software work? Is it a blinded study?

Page 4, line 91: What does "Inactivation of Kar2" mean?

The legends to figures need to be rewritten. In most cases, they contain more results and less experimental details. The legends should explain the figures, not the results.

How were RTG, SNF1 and HOG1 pathways blocked? No details are provided? Were these specific inhibitors?

As deletion strains were generated in-house, confirmatory tests should be included.

In fact, Methodology should include more details for each step.

Discussion is too long, repetitive at places and should be reduced. What does the sentence "a normal condition yeast cells experience outside of typical laboratory conditions" mean?

Version 1:

Reviewer comments:

Reviewer #1

(Remarks to the Author)

My concerns have been addressed and I favor publication.

Reviewer #2

(Remarks to the Author)

The authors have very thoroughly and adequately addressed all previous criticisms. The revisions have significantly strengthened the manuscript, and the presented findings are robust and compelling. I believe this work represents a valuable and impactful contribution that will advance the field.

Reviewer #3

(Remarks to the Author)

The authors have addressed my comments.

We thank the reviewers for excellent suggestions which greatly improve the rigor, relevance and readability of our manuscript. As per reviewers' feedback, we have rewritten the discussion, revised the figure legends and simplified the schematic describing our findings. We have additionally made the following organizational changes:

- i) Original Figure 5g is Figure 6a in the revised manuscript. The remaining panels in Figure 6 also comprise of data on human fibroblasts but generated during revision.
- ii) Original Figure 6 is Figure 7 in the revised version.
- iii) Original Figure 7, which depicted the model and crosstalk, has been removed. Instead, we have included a simpler schematic describing the major findings of our study in Figure 8f.
- iv) Original Figures 2d-e are Figures 8a-b, respectively, in the revised manuscript. Original Extended data Figures 2a-b are Extended data Figures 8a-8b, respectively, in the revised version. We believe that it was more suitable to group the above experiments on inositol deprivation along with revision work on mutants causing lipid bilayer stress (Figure 8c) or impairing lipid droplet biogenesis (Figures 8d-e). Thereby, Figure 8 comprises of data addressing the impact of lipid imbalance on peroxisomes. Consequently, the text pertaining to the data on inositol deprivation has been moved to the later portion of the results section of the revised manuscript.
- v) Flowcharts in Figures 2h and 4a in the original version have been slightly modified to more accurately depict the signaling pathways (Figures 2f and 4a, respectively, in the revised version).
- vi) The phrasing for some sentences has been modified for improved lucidity.
- vii) Additional details pertaining to original experiments, as well as new methods associated with revision experiments, have been added to the Methods section.

We have addressed all of reviewers' queries (bold) using point-by-point responses below and attached the data from revision experiments.

Reviewer #1 (Remarks to the Author)

This study demonstrates that misfolded protein stress in the ER induces peroxisome proliferation in yeast. The process is partially dependent on the TOR1 and heat shock proteins. There is an extensive analysis of whether stress induced peroxisome proliferation requires various kinases and transcription factors. These are fascinating findings since there was no previous evidence that peroxisome proliferation could be driven by unfolded protein in the ER. There are several ways the study could be improved.

1. Does peroxisome proliferation following ER stress help cells survive? For example, are *pex* mutants more sensitive to ER stress or do they recover from tunicamycin or DTT treatment more poorly than wt cells? Do mammalian cells that lack peroxisomes or that fail to proliferate them following ER stress have defects recovering from ER stress? More investigation of the function of stress induced peroxisome proliferation is necessary for this study to be appropriate for Nat Comm.

We have addressed this point by comparing the survival of *pex3Δ K. phaffii* cells (Figure 5f in the original and revised versions) and *pex16KO* primary human fibroblasts to their WT counterparts (Figure 6c in revised manuscript). Both yeast Pex3, and human Pex16 are essential for generating peroxisomes.

Extended data Figure 5b

Figure 5f

Previous work in *S. cerevisiae* has shown that WT cells pre-treated with a low dose of tunicamycin (0.5 μ g/ml) survive treatment with a higher dose (10 μ g/ml) compared to untreated cells indicating that the pre-treatment induces important adaptive changes for withstanding higher drug doses¹. Moreover, they also observed that loss of Pex34, a peroxisome membrane protein enriched at the peroxisome-mitochondrial contact site, reduced this adaptation and thereby compromised survival.

We performed a similar assay in *K. phaffii* where we compared the relative survival of WT and *pex3Δ K. phaffii* cells pre-treated with a low dose of (0.5 μ g/ml) of tunicamycin followed by a higher dose (10 μ g/ml). We observed that the *pex3Δ* strain, which is unable to generate peroxisomes has ~50% reduced survival compared to WT. Additionally, the impairment of *pex3Δ* cells in surviving ER stress is exacerbated with relative survival dropping to ~10% when glucose availability is limited, and cells are more heavily reliant on respiration.

Extended data Figure 6

Figure 6c

We also assessed the survival of WT and *pex16KO* primary human fibroblasts using a similar set up (Figure 6c in revised manuscript). For this, cells were subjected to a 12h-long primary treatment with 2-fold serially diluted concentrations of a 20 μ g/ml stock of tunicamycin, followed by a secondary treatment with 25 μ g/ml tunicamycin or DMSO for 12h. After this, viability was assessed using CellTiter-Blue assay. Pre-treatment (all tested concentrations) with tunicamycin enhanced the survival of WT cells exposed to 25 μ g/ml treatment. However, *pex16KO* human fibroblasts exhibited minimal adaptation corroborating with data from yeast experiments.

2. The results in Fig. 5g should be quantified. Please show higher mag images without the actin staining so it is easier to see the peroxisomes. Determining whether proliferation occurs in multiple cell types would strengthen the study.

Data from Figure 5g comprise Figure 6a in the revised version, and we have added high mag images of PMP70-marked peroxisomes now. The quantification for the number of peroxisomes per unit volume is presented in revised Figure 6b.

Figure 6a

Figure 6b

In this study, we have already shown that peroxisome proliferation is conserved in two different yeasts that diverged 250M years ago as well as primary human fibroblasts. We feel that extending this enquiry to other cell types is secondary to, and beyond the scope of this study. Importantly, it would not add to the message of this manuscript.

3. It would be good to know whether the amounts of tunicamycin and DTT used to cause ER stress are lethal to the cells. If they are, what fraction of the cells are viable after 5 hours of treatment? Please also show peroxisomes proliferation over time.

We quantified the % dead cells using propidium iodide (PI) staining following 5h-long treatment with tunicamycin or DTT. These data comprise Extended data Figure 1i (*S. cerevisiae*) and Extended data Figure 5a (*K. phaffii*) in the revised manuscript. In *S. cerevisiae*, these treatments cause minimal death with ~97% cells still alive, whereas in *K. phaffii*, ~63% cells survive the treatment.

We included a time course of peroxisome number as quantified by Pex3-GFP after treatment with tunicamycin (shown below) as Extended data Figure 1n.

4. Only one mutant is used to determine whether lipid stress induces peroxisome proliferation. Other mutant that significantly alter ER lipid composition could be tried such as *cho1* and *cho2*.

The pathway for phospholipid synthesis in *S. cerevisiae* (panel (a) ²) shows the genes involved in the biosynthesis of four major phospholipids in *S. cerevisiae*: PI, PS, PE and PC. In our manuscript, we induced lipid stress using inositol deprivation from the growth media (thereby impairing PI synthesis) and confirmed that it activates the UPR. Absence of inositol does not significantly increase the number of peroxisomes, and only mildly induces the heat shock response pathway in comparison to tunicamycin treatment (Figure 8a-b and Extended data Figure 8a-b in revised manuscript).

Extended data Figure 1n

As per the reviewer's suggestion, we altered ER lipid composition by blocking PC synthesis using *cho2Δ* and *opi3Δ* mutants, since unlike *cho1Δ*, these have been reported to induce lipid bilayer stress³.

These data are included in Figure 8c and depicted in (b) above, and show that the loss of Opi3, and to a lesser extent, of Cho2, increased peroxisome proliferation. However, unlike inositol deprivation, the loss of Opi3 or Cho2 also induces HSR⁴. It is therefore possible that the increased peroxisome proliferation observed in these mutants is not simply a consequence of altered lipid composition, but an indirect effect of impaired proteostasis-driven HSR activation.

5. The study shows that blocking pexophagy does not affect ER expansion, but what about blocking bulk autophagy?

Figure 7c

We presume that the reviewer meant the impact of blocking autophagy on peroxisome proliferation, and not on ER expansion. We performed this experiment and observed that blocking bulk autophagy did not cause any dramatic increase in the number of peroxisomes. This observation is expected since bulk autophagy is minimal during growth in glucose media. Notably, previous reports demonstrate that ER stress inhibits TOR, thereby activating autophagy, and this is corroborated by our own results (Extended data Figure 4). Importantly, our manuscript shows that

inhibition of TOR, by addition of rapamycin, mimics the effects of ER stress, thereby suggesting that peroxisome proliferation can proceed even with activation of autophagy, rather than its impairment.

6.The discussion is very long and includes a lot of material that seems better suited for a review or that could be said more concisely. The authors may want to consider shortening it.

As per the reviewer's suggestions, we have rewritten the discussion.

Reviewer #2 (Remarks to the Author)

In this manuscript, Shukla et al. describe experiments arguing that protein misfolding stress induces peroxisome proliferation. They exposed cells to various pharmacologic/chemical, genetic, and environmental perturbations and measured the number of peroxisomes in each case. While most of their experiments were done in budding yeast, they extended some of these studies to methylotrophic yeast cells and human fibroblasts. From all these experiments, the conclusion was that inputs leading to protein misfolding/stress would increase peroxisome copy number. The information likely involves the heat shock response pathways and an antagonistic role of the TOR pathway. Overall, the manuscript is interesting, and the experiments are done well. However, after reading the manuscript, the reader is left with the sense that the story is very descriptive and does not move the field significantly, given the well-established role of other perturbations that induce peroxisome biogenesis, such as oxidative stress.

1. The authors measure peroxisome copy number as a proxy for peroxisome biogenesis. Presumably, changes in the total peroxisomal compartment (i.e., the sum of all individual peroxisomes in each cell) more accurately reflect the relationship between stress and peroxisomes and the metabolic roles of these organelles. But they focus exclusively on copy numbers. Is the size of individual peroxisomes similar in all the conditions they test? Are there cases where there might be an increase in copy number (more but smaller peroxisomes) that does not significantly change the total peroxisomal volume per cell?

We addressed this query using peroxisome area measurements performed by perox-per-cell and included the data in Extended data Figures 1j-k.

The data from three replicates indicate that in response to tunicamycin treatment, while the average area of individual peroxisomes does not change significantly, the average cumulative area of all the peroxisomes per cell shows significant increases. This suggests that increased copy number accounts for the elevated cumulative peroxisomal area, viz. total peroxisomal compartment, per cell.

2. Is the response they observe separate from the oleate-induced peroxisome biogenesis? Their results with *dnm1* deletants argue that they are. But are the two responses additive? What happens if they add oleate on top of the various stressors they test?

Figure Only for reviewers. Not included in revised manuscript.

effects of tunicamycin and oleate are not synergistic. We think that these data do not add to the current message of our manuscript and hence have not included them in the revised manuscript.

We addressed this request by comparing the effect of DMSO or tunicamycin treatments on the number of peroxisomes in glucose- or oleate-grown *S. cerevisiae* cells. The data show that while tunicamycin increases peroxisome number in glucose-grown cells, it barely causes any further increase in peroxisome number of cells growing in oleate media.

This suggests that the

3. Related to their role in fatty acid metabolism, what happens to lipid droplets in these stresses, and how peroxisome biogenesis is affected in stressed and unstressed cells if lipid droplet homeostasis is perturbed genetically? Likewise, what happens if fatty acid synthesis is affected (e.g., by blocking fatty acid synthase with drugs, such as cerulenin, or by relevant genetic perturbations)?

To address this query as well as the Reviewer's Question 4, we generated a strain expressing GFP-ePTS1 (peroxisome marker) as well as Erg6-mCherry (lipid droplet marker). Tunicamycin-treatment caused a negligible increase in the number of lipid droplets (a) whereas the increase in peroxisomes was large (b) as quantified in the same cells. Md denotes median number of peroxisomes.

To examine the impact of tunicamycin on the contact sites between peroxisomes and lipid droplets, we quantified the fraction of peroxisomes that spatially overlapped with lipid droplets, considering an overlap to occur when at least one pixel of the peroxisome signal overlapped with the lipid droplet signal. We calculated this for each cell by normalizing the the number of GFP-ePTS1 puncta that overlapped with Erg6-mCherry puncta with the total number of peroxisome and lipid droplet puncta. We reasoned that cells with elevated number of puncta might show higher instances of overlap compared to cells containing relatively fewer puncta,

and since tunicamycin causes the cell population to shift towards exhibiting higher peroxisome numbers, examining pooled population data could lead to inaccurate conclusions. Therefore, we combined data from two replicates (shown above in Figures a-b) and binned cells into a matrix of lipid droplet vs peroxisome numbers as shown in (c-f) below and computed an average for all the cells within a bin. As seen in (c), tunicamycin increases the number of cells containing higher number of peroxisomes, whereas (d) shows the average fraction of peroxisomes that overlap with a lipid droplet in DMSO and tunicamycin-treatment conditions. We next subtracted the values for DMSO- from tunicamycin-treatment to generate a difference matrix as shown in (e) and calculated statistical significance for the treatments by applying the Mann-Whitney test to the matrix followed by FDR correction. These analyses indicated a negligible increase in the fraction of peroxisomes that spatially overlapped with lipid droplets. However, none of these increases were statistically significant following FDR correction. We strongly feel that these data do not add to the thrust of our paper, which is focused on peroxisome dynamics and therefore we have not included them in the revised manuscript, but show this to address the Reviewer's question.

d

Matrix for average fraction of peroxisomes that overlap with an LD

e

As suggested by the reviewer, we also examined how peroxisome proliferation in response to ER stress is coordinated with lipid droplet homeostasis and presented these data as Figures 8d-e in the revised manuscript. To investigate this, we used the quadruple deletion strain (Δ QD), in which four key genes involved in neutral lipid synthesis (*ARE1*, *ARE2*, *LRO1* and *DGA1*) were deleted. The absence of lipid droplets in this strain, confirmed in (g), results in the retention of the lipid droplet marker, Erg6, at the ER.

As shown in (h), upon impairing lipid droplet formation using this quadruple deletion (Δ QD), peroxisome number was comparable to that in WT in DMSO, as well as after tunicamycin treatments. Therefore, the absence of lipid droplets did not prevent the tunicamycin-dependent peroxisome proliferation. Since peroxisomes play a role in fatty acid β -oxidation and not in fatty acid synthesis, we think that testing the impact of blocking the latter is not pertinent to the focus of our manuscript.

4. Following up on the above, contact sites between peroxisomes and lipid droplets are known to be dynamic. How are these sites affected by the various stresses tested?
We have addressed this query in the response to Question 3.

5. Their summary Figure 7 is a typical 'hairball' signaling figure, where everything regulates everything. Is it possible to replace it with a simpler, more meaningful one?
We have replaced this Figure with a simpler schematic describing the major findings of our study in Figure 8f in the revised version.

Reviewer #3 (Remarks to the Author):

1. Kar2 activation has also been reported with a defect in Sec59 (James et al. 2019, 10.1038/s41598-019-51054-7). Looking at Sec61-Sec63 selectively provides only a partial picture. The authors need to study the role of Sec59.

Sec59 encodes dolichol kinase, which catalyzes the synthesis of dolichyl monophosphate (Dol-P). Loss of function mutants of Sec59 are impaired in N-glycosylation and exhibit elevated UPR, as well as chaperones, including Kar2. We induced ER stress using tunicamycin, which induces ER stress by blocking N-glycosylation. Since effects of *sec59* mutants and tunicamycin impinge on the same pathway leading to ER stress, we think this experiment is redundant to what we have already included in the manuscript.

2. Figure 1: An important control is missing. The authors need to study the distribution pattern of GFP alone in WT and mutant cells.

We presume that the aim of examining GFP alone in WT and mutant cells was to confirm that any increased puncta formation in our peroxisomal puncta quantification assay had not arisen due to aggregation of GFP itself. This is an important control since our experimental treatments invoke protein misfolding. We thus quantified the number of GFP-ePTS1 or Pex3-GFP puncta in cells where either peroxisomal import (*pex5* Δ) or peroxisome biogenesis (*pex19* Δ) is blocked, respectively. Pex5 is the cargo receptor which mediates the import of matrix proteins into the peroxisome lumen and in its absence, GFP-ePTS1 is cytosolic. Pex19 is a cytosolic receptor essential for biogenesis of peroxisomes and therefore in its absence, no peroxisomal puncta of Pex3-GFP are visible. The data shows that both markers are unable to locate to puncta even after treatment with a higher dose (2 μ g/ml) of tunicamycin. This indicates that the puncta that we observe using both markers in WT cells represent bona fide peroxisomes.

3. How does the developed software work? Is it a blinded study?

The perox-per-cell software⁵ performs 2D image-based segmentation of cells and peroxisomes using Z-projections of 3D image stacks, and quantifies peroxisomal features such as number, area and intensity of puncta on a cell-by-cell basis. The quantification of peroxisomes by this software was comparable to manual quantification performed by two independent human-counters. The human counters were given a set of blinded images of WT as well as mutants with reduced (*vps1* Δ) or higher (*pex30* Δ) peroxisome numbers.

4. Page 4, line 91: What does “Inactivation of Kar2” mean?

We used the *kar2-159* temperature sensitive mutant in this study. We have used “inactivation of Kar2” as being synonymous to “loss of Kar2 function” upon temperature shift of *kar2-159* to the restrictive temperature.

5. The legends to figures need to be rewritten. In most cases, they contain more results and less experimental details. The legends should explain the figures, not the results.

We have now revised the figure legends as per reviewer’s feedback. Additional details which are not in the figure legends are included in the Methods section.

6. How were RTG, SNF1 and HOG1 pathways blocked? No details are provided? Were these specific inhibitors?

We blocked the SNF1- and HOG-pathways using *snf1Δ* and *hog1Δ* deletion strains, respectively. We blocked RTG pathway by using single deletion mutants *rtg1Δ*, *rtg2Δ* and *rtg3Δ*. These are explained in the legend for Figure 4.

7. As deletion strains were generated in-house, confirmatory tests should be included. In fact, Methodology should include more details for each step.

As per the reviewer’s suggestion, the confirmation data for the deletion strains generated in-house (*dnm1Δ*, *inp2Δ* and *snf1Δ*) are included in the Supplemental data. We have also added the primer list and the details for generation of deletion constructs in the same file. The *msn2Δ* *msn4Δ* was generated by crossing single deletions; rest of the deletion strains were either obtained from the MATalpha haploid deletion library or from other labs where they have been part of previous publications.

8. Discussion is too long, repetitive at places and should be reduced. What does the sentence “a normal condition yeast cells experience outside of typical laboratory conditions” mean?

We have rewritten the discussion as per reviewer’s suggestion. The sentence referred by the reviewer from our original manuscript alluded to ER stress being one of the several kinds of stresses that are experienced by yeast cells during growth in their natural environments (for example, secretion of antibiotics from surrounding species in a community setting or sudden temperature changes resulting in proteotoxic stress). However, we have now removed that sentence from the revised manuscript.

References cited here

- 1 Hijazi, I., Wang, E., Orozco, M., Pelton, S. & Chang, A. Peroxisomal support of mitochondrial respiratory efficiency promotes ER stress survival. *J Cell Sci* 135 (2022). <https://doi.org/10.1242/jcs.259254>
- 2 Carman, G. M. & Han, G. S. Regulation of phospholipid synthesis in yeast. *J Lipid Res* 50 Suppl, S69-73 (2009). <https://doi.org/10.1194/jlr.R800043-JLR200>

- 3 Thibault, G. et al. The membrane stress response buffers lethal effects of lipid disequilibrium by reprogramming the protein homeostasis network. *Mol Cell* 48, 16-27 (2012). <https://doi.org:10.1016/j.molcel.2012.08.016>
- 4 Brandman, O. et al. A ribosome-bound quality control complex triggers degradation of nascent peptides and signals translation stress. *Cell* 151, 1042-1054 (2012). <https://doi.org:10.1016/j.cell.2012.10.044>
- 5 Neal, M. L. et al. Automated, image-based quantification of peroxisome characteristics with perox-per-cell. *Bioinformatics* 40 (2024). <https://doi.org:10.1093/bioinformatics/btae442>